# $100K or 100 Days: Trade-offs when Pre-Training with Academic Resources

**Apoorv Khandelwal, Tian Yun, Nihal V. Nayak, Jack Merullo, Stephen H. Bach, Chen Sun & Ellie Pavlick**
Department of Computer Science
Brown University
apoorvkh@brown.edu, ellie_pavlick@brown.edu

## Abstract

Pre-training is notoriously compute-intensive and academic researchers are notoriously under-resourced. It is, therefore, commonly assumed that academics can't pre-train models. In this paper, we seek to clarify this assumption. We first survey academic researchers to learn about their available compute and then empirically measure the time to replicate models on such resources. We introduce a benchmark to measure the time to pre-train models on given GPUs and also identify ideal settings for maximizing training speed. We run our benchmark on a range of models and academic GPUs, spending 2,000 GPU–hours on our experiments. Our results reveal a brighter picture for academic pre-training: for example, although Pythia-1B was originally trained on 64 GPUs for 3 days, we find it is also possible to replicate this model (with the same hyper-parameters) in 3x fewer GPU–days: i.e. on 4 GPUs in 18 days. We conclude with a cost–benefit analysis to help clarify the trade-offs between price and pre-training time. We believe our benchmark will help academic researchers conduct experiments that require training larger models on more data. We fully release our codebase at: https://github.com/apoorvkh/academic-pretraining.

## 1 Introduction

AI research today is dominated by large pre-trained language and vision models, which are notoriously compute-intensive to produce. Even smaller and older models have large training costs: Pythia-1B required 64 GPUs for 3 days (Biderman et al., 2023) and Roberta required 1,000 GPUs for 1 day (Liu et al., 2019). A ubiquitous complaint among academic labs is that research on new architectures, data diets, and training procedures is prohibitively and increasingly expensive. As a result, groups often opt out of conducting controlled experiments involving pre-training.

Such decisions come at a cost to the field as a whole. Our research progress would be more competitive and diverse if the many smaller groups could also conduct these experiments. Using an example from computer vision: although the significant gains of the CLIP model (Radford et al., 2021) were initially credited to its contrastive loss, more controlled experiments later revealed that this was an over-attribution (Tschannen et al., 2023). The three-year lag between the initial, successful model and the more principled follow-up studies arguably led the field to *exploit* too soon, rather than continually *explore* more architectures and training objectives.

While academic research labs are the ideal place to pursue such principled analyses, in practice, there is little common knowledge about the costs of pre-training in academia. Given certain GPUs, how many days would pre-training take? Which types of models could we train, if we really wanted to, and which are unambiguously unattainable? This lack

---

* Our paper title "$100k or 100 days" is drawn from the stark comparison (Sec. 4.1) between replicating Pythia-1B with four H100 (8 days, $130k) or two 3090 GPUs (108 days, $4k).

of transparency is, itself, a barrier to academic research. It makes it difficult for students and their PIs to pursue more ambitious experiments, propose grants with realistic budgets, and prioritize their time. A clearer understanding about the resources that are available to our community—and the optimizations that can be made in pre-training—can enable us to make more informed decisions about where to focus our funding and intellectual energy.

In this paper, we seek to provide transparency about the feasibility of pre-training models given the current state of academic hardware. In all:

1. We conduct a survey and learn a common range for academic compute: namely, 1–8 GPUs that can be used for days (typically) or weeks (on the higher-end) at a time.

2. We empirically measure and report the time necessary to replicate several models on academic GPU configurations. We optimize performance by searching a space of efficient training methods. To develop the insights in our paper, we benchmark nearly 3,000 configurations in our search space and spend approximately 2,000 GPU–hours to do so.

3. We find that we are able to pre-train models using our optimizations and current hardware with 3x less compute than original reports. For example, Pythia-1B can be trained in 18 days on 4 A100 GPUs (i.e. 72 GPU–days, rather than 192). In many cases, our optimizations even enable training experiments that are otherwise entirely infeasible.

4. We conduct a cost–benefit analysis to help determine which hardware is best (enabling the fastest pre-training) given a certain financial budget. For example, a machine with those 4 A100 GPUs will cost $85k. It might be more effective to buy 2 H100 GPUs at $60k (which can train Pythia-1B in the same time).

We fully release our codebase and artifacts—which can be easily extended to test new models, training methods, and GPUs—so that our procedure can be used to benchmark more custom hardware and model configurations. Our benchmark is inexpensive to run and can result in large reductions in training time (or enable training entirely), especially on few GPUs.

## 2 What is "Academic Hardware"?

It's generally agreed that conducting NLP research is increasingly challenging on "academic hardware", but there is no single definition of what academic hardware is. As researchers, we often are in the dark about the type of resources we should have and what our peers have.

We conducted a survey asking AI researchers in academia about their compute budgets for research experiments. We shared our survey for 3 weeks in April 2024 by word-of-mouth, Twitter, and academic (inter-university) Slack channels. This procedure yielded responses from 50 researchers across 35 international institutions. PhD students accounted for the majority (approx. 60%) of our respondents.

Our survey asks about resources available for an individual or per-project basis, rather than for measurements aggregated across entire labs or departments. We report some of our findings here, and include the full list of survey questions, respondent demographics, and remaining analyses in Appendix A.

**Availability of GPUs.** While cloud compute is often proposed as a flexible way for academics to scale their compute resources, we find that, in practice, very few actually use this. In fact, the vast majority (85%) of respondents reported zero budget for cloud compute. Instead, academic researchers typically rely on on-premises compute provided by their institution. Owning hardware simply tends to be more cost-effective than renting: e.g. it costs less than $200K to own an 8 x 80 GB A100 machine, but ~$650K to reserve on AWS (p4de.24xlarge) for 5 years.

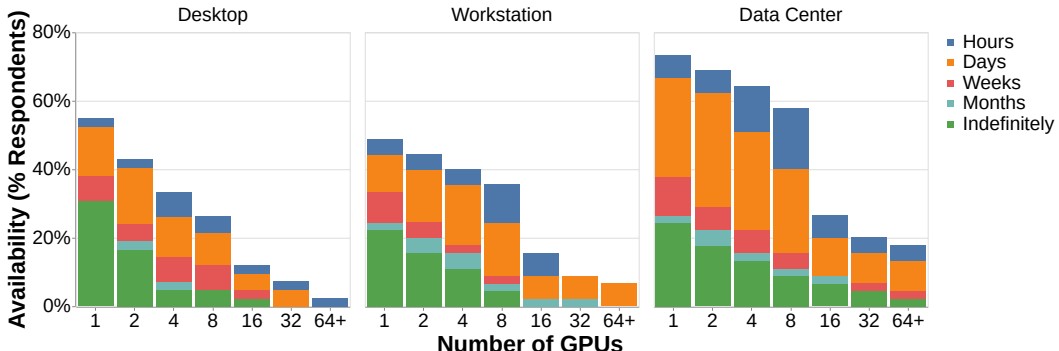

Figure 1: Per-experiment availability of GPUs (typically 24 GB RTX 3090s, 48 GB A6000s, and 80 GB A100s). Showing how long (at most) respondents can use N GPUs for.

That said, most academics are not satisfied with the compute provided by their institution. 66% of respondents rated their satisfaction with their compute clusters at less than or equal to 3 out of 5 (indicating that some desired experiments are prohibitively expensive). These trends are similar to those reported in Lee et al. (2023).

Several respondents mentioned a particular dissatisfaction with long wait times for allocating GPUs and limited connectivity between nodes. One respondent said the wait times can "sometimes be up to 2 to 3 days" and another emphasized they are "a lot longer during deadlines". 41% of respondents stated that they had no multi-node support in their institute cluster. Even if universities have multiple nodes, "there is no inter-node connectivity" for some, "which is a bummer for fitting any greater than 7B-parameter models".

We further surveyed our respondents' resources, measuring both the number and types (i.e. *Desktop*, *Workstation*, and *Data Center*) of GPUs available to them. This distinction of types is important, because, among other specifications, they most prominently differ in price, memory capacity, and processing power. For example, considering the most popular GPU models reported, the RTX 3090 Desktop GPU costs $1,300 today and features 24 GB memory and 70 TFLOPs/sec of 16-bit compute. The A6000 Workstation GPU costs $4,800 with 48 GB memory and 150 TFLOPs/sec in compute. And, the A100 Data Center GPU costs $19,000 with 80 GB memory and 310 TFLOPs/sec in compute.

In Fig. 1, we report the amount of time for which researchers can allocate these types of GPUs per experiment. In general, we find that the typical academic researcher can use 4 GPUs for days at a time. And, at the higher-end, some academics can use 4 for weeks and 8–16 for days at a time. 50% more respondents reported access to (the more expensive) Data Center GPUs than to Desktop or Workstation GPUs.

We also notice a disparity in resources regionally and by type of institution. A respondent from Oman said "finding GPUs in the Middle East is challenging". At the other extreme, a Swiss respondent mentioned there will soon be "[10,000][1] H100 GPUs for all Swiss universities under the SwissAI initiative". Finally, a professor at an American liberal arts college elaborated on their particular case: their institution did not have a compute cluster at all and instead had 4 RTX 3090 GPUs in a desktop.

**Use cases.** Most academics (70–80%) reported using GPUs for purposes such as fine-tuning, inference, and training small models. 57% also report that they use GPUs for model analysis. 17% report running pre-training experiments for models with less than 1B parameters. Based on our poll on user satisfaction, the majority of respondents want to and indeed would run more expensive types of experiments, if only they had the hardware for it.

---

[1]Respondent originally reported "1,000", but we found the quantity was actually 10,000 upon further verification (https://www.swiss-ai.org).

| Model | Size | RTX 3090 (24 GB) | | | | A6000 (48 GB) | | | | A100 (80 GB) | | | | H100 (80 GB) | | | |
|---|---|---|---|---|---|---|---|---|---|---|---|---|---|---|---|---|---|
| | | 1 | 2 | 4 | 8 | 1 | 2 | 4 | 8 | 1 | 2 | 4 | 8 | 1 | 2 | 4 | 8 |
| Pythia | 160M | 41 | 18 | 6 | 3 | 29 | 15 | 7 | 4 | 14 | 7 | 3 | 2 | 7 | 4 | 2 | 1 |
| | 410M | 151 | 69 | 35 | 19 | 105 | 49 | 26 | 15 | 50 | 25 | 12 | 6 | 25 | 13 | 6 | 3 |
| | 1B | 370 | 109 | 77 | 30 | 152 | 72 | 40 | 22 | 72 | 36 | 18 | 9 | 34 | 16 | 8 | 4 |
| | 2.8B | 1515 | 1040 | 485 | 177 | 934 | 292 | 150 | 88 | 342 | 148 | 71 | 35 | 166 | 77 | 31 | 15 |
| | 6.9B | — | 7157 | 1769 | 1250 | — | 1110 | 819 | 264 | — | 488 | 170 | 77 | — | 220 | 69 | 32 |
| RoBERTa | 350M | 1070 | 559 | 266 | 170 | 826 | 423 | 213 | 114 | 394 | 197 | 100 | 50 | 175 | 88 | 44 | 23 |
| Mamba | 2.8B | 1992 | 1217 | 444 | 304 | 1414 | 483 | 259 | 193 | 500 | 263 | 133 | 67 | 277 | 145 | 74 | 37 |
| ConvNeXt | 390M | 154 | 133 | 49 | 27 | 168 | 68 | 33 | 22 | 59 | 30 | 16 | 8 | 31 | 16 | 8 | 5 |
| ViT | 325M | 156 | 87 | 45 | 29 | 111 | 56 | 31 | 16 | 53 | 26 | 14 | 7 | 27 | 14 | 7 | 4 |

Table 1: Empirical training times (in days) for model–GPU combinations using optimal settings discovered in Sec. 3.2. "—" indicates infeasibility with all efficient methods.

**Takeaways.** Our survey suggests a common range for what constitutes "academic hardware" today: 1–8 GPUs—especially RTX 3090s, A6000s, and A100s—for days (typically) or weeks (at the higher-end) at a time. 10% of our respondents also report access to H100 GPUs: i.e. the newest-generation Data Center GPUs. In the following section, we investigate how long training takes on several types of GPUs and use this range for "academic hardware" in our investigations.

# 3 Measuring Training Time

Although we know some model and GPU properties (such as floating-point operations, processing power, etc.), these metrics are not commonsense for practitioners. A simple open question remains: how long will it take to train my model on my compute?

In this section, we determine the time to pre-train (and replicate) several models on different academic GPU configurations. We report optimal training times for all tested models and GPUs in Table 1 (and corresponding configurations in Appendix J). We provide further results in Sec. 4.

We use analytic (Sec. 3.1) and empirical (Sec. 3.2) approaches in our investigation. In our empirical approach, we test both naive training settings, as well as with combinations of efficient training methods (Sec. 3.2.1), to identify optimal configurations with minimal training time.

We analyze several well-known language and vision models (with between 100M–7B parameters): Transformer encoders [Roberta (Liu et al., 2019)] and decoders [Pythia (Biderman et al., 2023)]; state space models [Mamba (Gu & Dao, 2023)]; convolutional networks [ConvNeXt (Liu et al., 2022)]; and vision transformers [ViT (Dosovitskiy et al., 2021)]. In particular, these models report sufficient training details (architecture and hyper-parameters; Appendix C) for us to emulate replication (i.e. training with the same batch size, for the same number of steps, etc.). We use these original hyper-parameters whenever possible. Because we focus on characterizing which types of experiments are generally accessible to the average academic, we exclude any efficiency method that changes the training recipe (and that could change convergence) or requires extensive implementation-level modifications.

## 3.1 Analytically Inferring Training Time

As a first step, we use basic measurements to determine training time. That is, just using simple model properties and what we know about hardware from manufacturers, how long would pre-training take on various realistic academic setups?

We estimate pre-training time based on total training compute (FLOPs) for models and throughput (FLOPs/sec) for GPUs (and list these in Appendix E). We count the FLOPs for a model in one training step (i.e. forward and backward pass, given a single-element input batch), and linearly extrapolate by batch size and training steps to determine the total training FLOPs. We use GPU throughput values from hardware specifications and assume

the precision of the original model. Finally, we can directly infer training time from these FLOPs and throughput (FLOPs/sec) measurements.

This inference is an estimate, because existing automatic methods for counting FLOPs are approximators (e.g. counting only matrix multiplication operations). FLOPs also do not account for parameter update time. Finally, this is a highly simplified model of GPUs and neural networks: other factors create training bottlenecks, such as memory bandwidths and caches of GPUs, and architectures and operations of models. By ignoring these bottlenecks, we unrealistically assume 100% utilization of GPU cores. A true measurement becomes infeasible using a simple analytic approach, necessitating our empirical approaches (below).

### 3.2 Empirically Measuring Training Time

Our empirical measurement involves three steps:

1. **Maximum batch size.** We identify the largest batch size (by power of two) that will fit in GPU memory—approx. maximizing throughput (NVIDIA, 2023)—by incrementing the batch size (and computing a training step) until we run out of memory. We limit this value to our desired batch size for replication (Appendix C).

2. **Training step time.** We compute and measure training steps using the maximum batch size from (1). We note that this maximum size may be smaller than the desired value for replication. We perfectly compensate using gradient accumulation: i.e. by computing a commensurate number (N = desired/maximum) of forward/backward passes and accumulating their gradients between parameter updates. We measure the durations of 3 such forward/backward passes and 3 parameter updates, and extrapolate to the duration of a training step: i.e. for N passes and 1 update. We also run one additional pass and update ahead of this measurement as a warm-up.

3. **Overall training time.** We linearly extrapolate the training step time from (2) by the total number of training steps (Appendix C) to estimate the overall training time.

We first benchmark naive settings: i.e. we load and train models with their replication hyper-parameters and in an entirely off-the-shelf/default manner. We then investigate combinations of several efficiency methods (Sec. 3.2.1) to further optimize training speed. We use randomly generated input data (e.g. language tokens or image pixels) for our benchmark. We include further details about our software implementation and hardware specifications in Appendix B.

### 3.2.1 Efficient Training Methods

We consider several optimizations, which we categorize as free-lunch or memory-saving methods (described below). Free-lunch methods are strictly beneficial. In contrast, memory-saving methods may have multiple options, and entail up to 22 combinations (Appendix F) in our experiments. It is thus more difficult to select a combination of such methods that is guarenteed to improve (or at least not hurt) training time. We benchmark the resulting search space (i.e. combinations of memory-saving methods, with free-lunch methods always enabled) and report the minimum training time in our results.

**Free-lunch methods.** These methods improve throughput and might reduce memory consumption, with no cost to the end-user. (Of course, this "lunch" actually comes at the cost of several years of deep learning software and hardware R&D.)

1. We **compile** the model (Ansel et al., 2024): automatically building GPU-optimized, low-level (Triton) code from high-level (PyTorch) operations.

2. We use off-the-shelf **custom kernels** as drop-in replacements for PyTorch modules. These kernels are low-level implementations of neural network layers (in Triton or CUDA) that are handwritten for performance improvements beyond what a

compiler can offer. At the time of our experiments, we primarily use FlashAttention (Dao et al., 2022; Dao, 2024) in our Transformer models and SSM-specific kernels for Mamba (Gu & Dao, 2023).[2]

3. We perform 32-bit matrix multiplications and convolutions in **TF32 mode** (Stosic, 2020): ignoring the least significant 13 bits of 32-bit floating-point numbers internally during scalar multiplications (but not accumulations). This results in sizable improvements to throughput with no effect on convergence.

**Memory-saving methods.** The following methods reduce memory consumption at a cost to speed.

1. **Activation checkpointing** (Beaumont et al., 2019–2024): One can save just a subset of activations (i.e. checkpoints) during the forward pass (saving memory) and re-compute missing activations from the nearest checkpoint and as needed during the backward pass (spending compute).

2. **Model sharding** (Rajbhandari et al., 2019; Zhao et al., 2023): Typically, a full copy of all model states (including weights, gradients, and optimizer states) are stored on each GPU. Instead, one can eliminate redundant copies by dividing (or "sharding") these states among the GPUs. Then, when a state is needed during training but missing from a GPU, that state will be temporarily copied from its corresponding GPU. Although this method can dramatically save memory on each GPU, it can also incur a cost via communication time.

3. **Offloading** (Rajbhandari et al., 2019; Ren et al., 2021): To conserve even more GPU memory, one can offload optimizer states and model parameters into system memory (RAM). These states will be communicated to the GPUs as needed. This method can incur a large communication cost due to low RAM–GPU bandwidth.

## 4 Results

We measure pre-training cost in training time (days) and follow the recommendations of Dehghani et al. (2022) to contextualize our results with respect to hardware and training settings. That said, some settings are entirely *infeasible*, e.g. due to excessive system or GPU memory consumption, even at batch size 1. We report optimal training times from our empirical approach for all models and GPUs in Table 1. We show naive training times and deaggregated results (e.g. for all models and GPU counts) in Appendix G.

We first consider the results of our naive empirical baseline for Pythia models. We find that the analytic inferences from Fig. 2a are overly optimistic by a factor of 6×, underscoring the importance of empirical benchmarks for realistic estimates. Moreover, in Fig. 2b, we can see that 9 of the 20 model–GPU configurations are entirely infeasible. As a reference point, we find that it would take 41 days to train Pythia-1B from scratch on 4 A100 GPUs (Appendix G)—notably higher than the compute budget (i.e. days or weeks) that is afforded by academic researchers (Sec. 2).

By finding an *optimal* training configuration (with the efficient methods in Sec. 3.2.1), we see 4.3× average training time speedups compared to the naive baseline (Fig. 2a). Now, our 4 A100 x Pythia-1B example only requires 18 days (Table 1) instead of 41, which falls within the means (i.e. weeks) of our better-resourced survey respondents. With >1 GPU, all 20 model–GPU combinations become feasible using optimal methods (Fig. 2b).

In Fig. 3, we can see that the optimal combination of memory-saving methods often results in training time gains over using free-lunch methods alone (even up to 71%), especially for GPUs with less memory (e.g. RTX 3090) or for larger models. This is surprising—because memory-saving methods are advertised to strictly reduce training speeds—but occurs because the saved memory can sometimes be repurposed for larger batch sizes, inducing

---

[2]FlexAttention (He et al., 2024) and the Liger Kernel collection (Hsu et al., 2024) are also prominent, more recent examples.

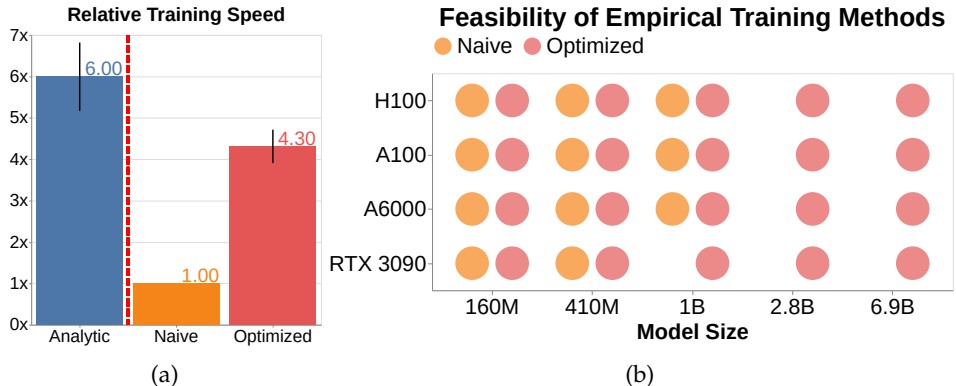

Figure 2: (a) Comparing the average speedup of our optimized training settings over the naive empirical baseline. Results are shown for Pythia models where naive setting is *feasible*. Analytic method included for reference. Error bars indicate confidence intervals. (b) Indicating which GPU–model combinations the naive and optimized settings are feasible in. For Pythia models and where >1 GPU is in use.

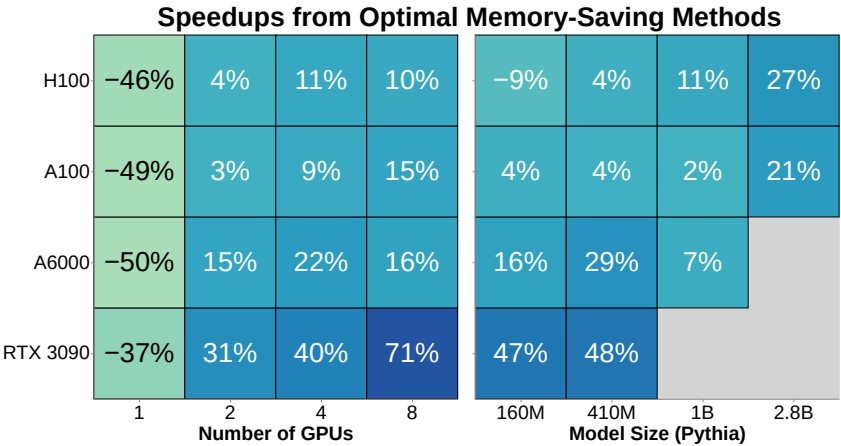

Figure 3: Average training speedups for Pythia models by using optimal memory-saving methods, in addition to free-lunch methods alone. Higher is better. By number of GPUs (top) and model size for >1 GPU (bottom). Gray indicates infeasibility with free-lunch methods alone. Only comparing settings where free-lunch methods are feasible. We can see that using the optimal memory-saving methods from our search space can offer up to 71% reduction in training time (compared to using no memory-saving methods) in some settings.

higher throughput. That said: when using only 1 GPU (when model sharding is a no-op), all memory-saving methods are detrimental to throughput for Pythia models, so these are likely particularly advantageous when free-lunch methods alone are infeasible.

We observe the importance of benchmarking all combinations of memory-saving methods in Fig. 4: on average, the optimal (or "best") discovered configuration is 1.3x faster than the second best, 2.3x faster than the median, and 4.7x faster than the worst-case configurations. So we find significance in identifying the most optimal configuration (rather than the second best) and also expect training to take twice as long using an un-informed, arbitrary configuration! The median case is also 1.7x slower than using free-lunch methods alone, so there are rather few combinations that indeed result in gains, rather than significant degradations. In all, we find that this exhaustive search is relatively inexpensive (e.g. a few hours per model–GPU setting) compared to the prospective additional costs to training time (e.g. many days) by picking an arbitrary configuration.

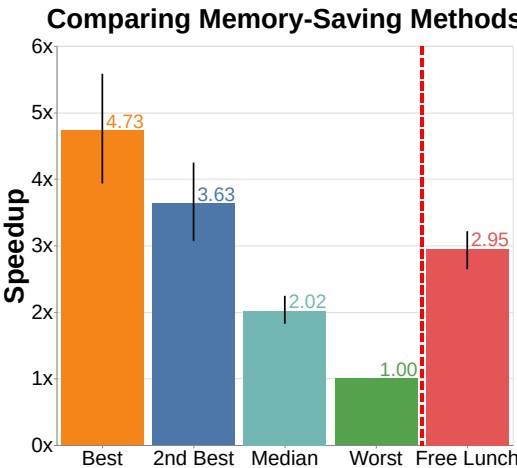

Figure 4: Comparing average training speedups among combinations of memory-saving methods for Pythia models in the >1 GPU case. "Free Lunch" as reference where no memory-saving methods are enabled. Higher is better. Only comparing settings where free-lunch methods are feasible. Error bars indicate confidence intervals. As in Fig. 3, we show gains from using the optimal memory-saving method over no memory-saving methods (Free Lunch). Here, we also show degradations from the median selection of methods, which we would expect if randomly choosing methods.

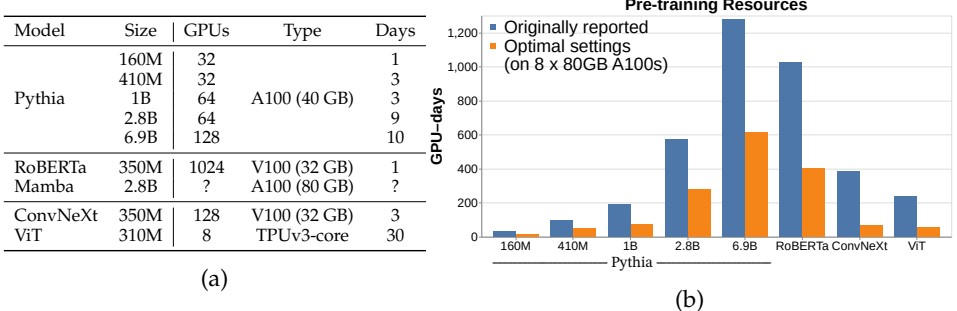

| Model | Size | GPUs | Type | Days |
|---|---|---|---|---|
| Pythia | 160M | 32 | A100 (40 GB) | 1 |
| | 410M | 32 | | 3 |
| | 1B | 64 | | 3 |
| | 2.8B | 64 | | 9 |
| | 6.9B | 128 | | 10 |
| RoBERTa | 350M | 1024 | V100 (32 GB) | 1 |
| Mamba | 2.8B | ? | A100 (80 GB) | ? |
| ConvNeXt | 350M | 128 | V100 (32 GB) | 3 |
| ViT | 310M | 8 | TPUv3-core | 30 |

(a)

(b)

Figure 5: (a) Resources used to train original models (inferred from the respective papers or Github repositories). (b) Compute (GPU–days) used to originally train models vs. with our discovered optimizations on current hardware (8 x 80 GB A100 GPUs). Lower is better.

In Fig. 5, we compare the total compute needed to originally pre-train these models with that for our optimal settings on 8 A100 GPUs. We find that our setting consumes 3.0× less compute on average (in GPU–days with "Data Center" GPUs).

## 4.1 Cost–benefit analysis

In Fig. 6, we consider the time to train the Pythia-1B model on our different GPU configurations. We compare overall hardware costs and single experiment costs (i.e. the training time-normalized cost, conservatively assuming a five-year hardware lifespan). For example, if our budget is $40k, we might opt to purchase a machine with 8 RTX 3090 GPUs, which can train Pythia-1B in 30 days (i.e. the least amount of time among hardware in our budget). If we have a more competitive budget: it would surprisingly be more cost-effective to buy 4 H100 GPUs ($130k) than 8 A100 GPUs ($160k)—even with half the memory—as both train this model in 8–9 days. We can see this trend between A100 and H100 GPUs when comparing costs per experiment: a training run will cost $800 if using A100s, but $600 if using H100s. That said, each H100 ($30k) plainly costs more than an A100 GPU ($19k). If

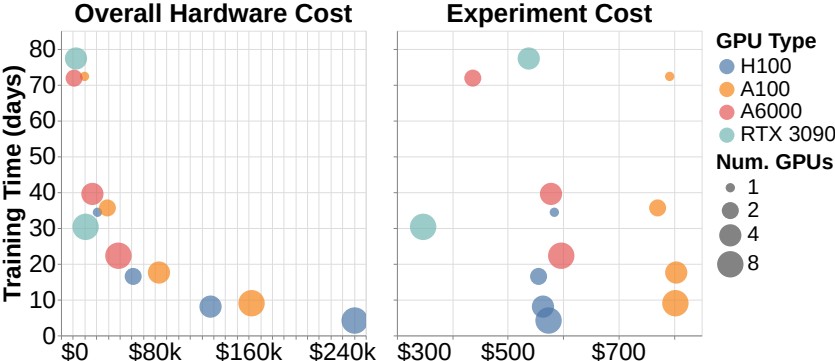

Figure 6: Hardware cost vs. pre-training time for Pythia-1B. Overall cost (top); experiment-normalized cost (bottom). We truncate a few outliers for clarity here and show the expanded plots with log scale in Appendix I.

our budget knows no bounds, a machine with 8 H100 GPUs ($250k) would be ideal, as it can train Pythia-1B faster than all other configurations: in a mere 4 days.

We perform this analysis using current hardware quotes at the time of writing and provide a breakdown in Appendix H. We assume a price per GPU ($1,300 per RTX 3090; $4,800 per A6000; $19k per A100; $30k per H100). We additionally consider system costs (varying by the number of GPUs a machine can support). For example, a $1,000 consumer/desktop machine can run any single GPU listed above. A desktop that supports 2 GPUs (with more compute/memory and a larger power supply) will cost $1,500. However, 4 GPUs and 8 GPUs need substantially more resources: we must instead use GPU servers (respectively priced at $7,500 and $10,500).

## 5 Related Work

**Compute surveys.**  Lee et al. (2023) surveys the NLP community (including both academia and industry) and investigates respondents' access to compute resources and downstream effects on the environment and the reviewing process. To the best of our knowledge, this is the only prior work to conduct a compute survey in AI or GPU-centric domains. In contrast, we survey the broader AI community and specifically focus on academic access to compute. Our compute survey not only asks how many GPUs are available, but also which type and for how long may they be used at a time—distinctions that influence the scope of possible experiments.

**Training recommendations.**  To improve training speed, the conventional wisdom (Hugging Face, 2024; Godbole et al., 2023; Bekman, 2023–2024) is to *tune-by-hand* using specialized knowledge to select efficiency methods. Our codebase abstracts away these details: we help find efficient settings as long as the end-user can define their model (including hyper-parameters, inputs, and objectives). We propose an automatic search to do so.

Concurrent work (Hagemann et al., 2024) similarly searches among efficiency methods, but targets the large model, many GPU regime and is specific to the LLAMA architecture. Deepspeed Autotune (Rasley et al., 2020) is another automatic search method that focuses specifically on batch size, Zero-based model sharding, and Zero-specific settings. Both works report metrics, such as model FLOPs utilization (Chowdhery et al., 2023) and throughput, which can be used for comparing methods, but are not as practical as training time.

**Training simulation.**  Rather than by taking empirical measurements, several recent works (Hu et al., 2022; Lu et al., 2023; Won et al., 2023; Duan et al., 2024; Wang et al., 2025; Feng et al., 2024) estimate training time via simulation: i.e. by tracing the primitive operations (or execution graph) entailed by a training workload and modeling runtime with respect

to hardware resources (e.g. GPU memory and throughput, network topology and latency, and so forth). The ideal simulator would be fast, accurate, and require fewer resources at simulation time (e.g. 1 GPU or node vs. a large-scale cluster of nodes). However, this is an ongoing line of research: even the most recent work (Feng et al., 2024) has incomplete support for training optimizations (e.g. Zero-based parallelism) and reports an 8% error rate. Prior works require the full cluster and report 20–40% error rates. And, these works generally require complex, low-level implementations.

On the other hand, our work opts to measure training time empirically. In doing so, it is simple, accurate, and has extensive support for possible training optimizations. Although this method requires the full cluster of nodes for 2 hours (on average), we believe this is a relatively inexpensive cost compared to the overall pre-training time (e.g. 7–30 days) and expected savings (40%; Fig. 4).

**Efficient pre-training recipes.**    Several works propose "recipes" to pre-train certain models in a short time on few GPUs. Izsak et al. (2021); Geiping & Goldstein (2023); Portes et al. (2023); Sanyal et al. (2024) pre-train BERT variants for between 1 hour and 1 day on 1–8 GPUs. However, these works achieve their speedups by: making large architectural modifications, altering the training objective, reducing precision, changing the optimizer and hyper-parameters, and so forth. These works do not replicate BERT, but actually pre-train variants to sufficient performances. Nawrot (2023) pre-trains T5-Base in 1 day on 1 A100 GPU with similar changes, but without modifying the model architecture and with a larger emphasis on efficiency methods, such as those in Sec. 3.2. Sehwag et al. (2024) trains a diffusion transformer in 3 days on 8 H100 GPUs by proposing new data masking strategies and using additional synthetic data.

Unlike these works, our paper indeed assumes replication settings: we make no modifications that affect a known model or its training recipe. Thus, our approach is entirely model-agnostic and can be flexibly extended to any deep learning model. The problem of finding good model architectures and training recipes (e.g. via hyper-parameter search) is orthogonal to conducting the controlled pre-training studies this paper aims to enable.

**Hardware recommendations.**    Dettmers (2018; 2023) offers extensive details on purchasing and using hardware for deep learning. On the other hand, our paper gives simplified recommendations, but emphasizes comparing hardware by training time (rather than by throughput).

# 6    Conclusion

The gulf between industry and academic compute has been clear for some time and is growing. There even appears to be an expertise gap related to larger-scale training. And so, it is unsurprising that few academic researchers train models from scratch.

In this paper, we provide insights into the current state of academic compute and the feasibility of pre-training on this hardware. We show that, in some cases, it is indeed possible for motivated academic groups to train models with billions of parameters. Our codebase and benchmark can be easily used to determine ideal settings for training models on one's hardware. Our benchmark can be run on short allocations of cloud compute before one makes large investments in hardware. We hope to help the academic community begin training larger models on more data, so that they can be more closely involved in the science of building new deep learning models.

**Limitations**

Although our codebase supports the multi-node setting, we only conduct single-node experiments (as this is the most common academic setup). Our codebase is an abstraction over prominent deep learning libraries, especially: PyTorch (Paszke et al., 2019), Hugging Face Transformers (Wolf et al., 2020), and Deepspeed (Rasley et al., 2020). The specific combination of model–hardware–efficiency settings can be complex. We defer all setting-

specific training failures (e.g. compilation or sharding failures for a certain model on a certain GPU; or other abnormal behaviors) to their respective upstream libraries. Certain efficiency methods can only be used on (somewhat) recent hardware (Appendix F). Actual training runs may see additional overhead from the training loop. Our reported results are functions of our exact hardware configurations (Appendix B). Finally, perfectly replicating a model is hard—very few works [e.g. (Biderman et al., 2023; Groeneveld et al., 2024)] *fully* open-source their data and models—so we defer replication failures to original works.

**Reproducibility**

We make our code fully available so that all of our experiments can be exactly replicated (within the error bounds of hardware and trial variance). We provide a complete, versioned list of our dependencies, such that our software environment can be reproduced exactly. We provide anonymized survey results and all results from our experiments such that further analyses may be conducted.

**Acknowledgments**

We are immensely thankful for the respondents to our academic compute survey. We would also like to thank Akshat Shrivastava, Alexander Koller, Chris Callison-Burch, Francisco Piedrahita Velez, Jennifer Hu, Michael A. Lepori, Nate Gillman, and Zheng-Xin Yong for their generous feedback on this work. This research was conducted using computational resources and services at the Center for Computation and Visualization, Brown University. This work was partially supported by a Brown University Presidential Fellowship for A.K. Disclosure: S.B. is an advisor to Snorkel AI, a company that provides software and services for data-centric artificial intelligence.

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

# A  Academic Compute Survey

---

**Q1. What is the name of your institution?**
**Q2. What is your role?** Answer: Undergraduate Student, Master's Student, Ph.D. Student, Postdoc, Professor, [Other]
**Q3. How would you categorize your research area(s)?** Answers: Machine Learning, Deep Learning, Natural Language Processing, Computer Vision, Reinforcement Learning, [Other]

---

**Q4. What do you use GPUs for?** Answers: Training (with small data / models), Pre-training (with large data / models), Fine-tuning, Inference, Model Analysis, Rendering, [Other]
**Q5. How satisfied are you by your current access to GPUs? (1–5)**
Answer: **(1)** "I cannot run most experiments that I wish to" **(2)** "I can run some experiments, but often experience difficulties (e.g. long wait times)" **(3)** "I can run many but not all experiments that I wish to" **(4)** "I face some difficulties (e.g. long wait times) with my most expensive experiments" **(5)** "I have no trouble running any experiments"
**Q6. What is your monthly budget (in USD) for cloud compute (e.g. AWS, GCP, etc)?**

---

**Q7. Which Desktop GPUs are you able to use at your university's compute cluster?**
**Answers**: Not sure, None available, Pascal (GTX 1000 series / Titan X Pascal / Titan Xp), Volta (Titan V), Turing (GTX 1600 series / RTX 2000 series / Titan RTX), Ampere (RTX 3000 series), Lovelace (RTX 4000 series), [AMD] Radeon VII, [AMD] Radeon RX 7900 XT / XTX, [Other]
**Q8. What is the largest memory Desktop GPU typically available to you?**
Answer: Not sure, None available, 4 GB, 8 GB, 12 GB, 16 GB, 20 GB, 24 GB, 32 GB
**Q9. How long can you use these Desktop GPUs for?** On an individual or per-project basis. Select 1 option per row.
Answers (Rows): 1 GPU, 2 GPUs, 4 GPUs, 8 GPUs, 16 GPUs, 32 GPUs, 64+ GPUs
Answers (Columns): N/A, Hours, Days, Weeks, Months, Indefinitely

---

**Q10. Which Workstation GPUs are you able to use at your university's compute cluster?**
Answers: Not sure, None available, Turing (Quadro RTX series), Ampere (RTX A2000 / A4000 / ... / A6000), Lovelace (RTX 4000 Ada / 5000 Ada / ... / 6000 Ada), [AMD] Radeon PRO VII, [AMD] Radeon PRO W6800, [AMD] Radeon PRO W7800 / W7900, [Other]
**Q11. What is the largest memory Workstation GPU typically available to you?**
Answer: Not sure, None available, 16 GB, 24 GB, 32 GB, 48 GB
**Q12. How long can you use these Workstation GPUs for?** On an individual or per-project basis. Select 1 option per row.
Answers (Rows): 1 GPU, 2 GPUs, 4 GPUs, 8 GPUs, 16 GPUs, 32 GPUs, 64+ GPUs
Answers (Columns): N/A, Hours, Days, Weeks, Months, Indefinitely

---

**Q13. Which Data Center GPUs are you able to use at your university's compute cluster?**
Answers: Not sure, None available, Pascal (P4 / P40 / P100), Volta (V100), Turing (T4), Ampere (A2 / A10 / A30 / A40 / A100), Lovelace (L4 / L40), Hopper (H100 / H200 / GH200), [AMD] Radeon PRO V620, [AMD] Instinct MI, [Other]
**Q14. What is the largest memory Data Center GPU typically available to you?**
Answer: Not sure, None available, 16 GB, 24 GB, 32 GB, 48 GB, 64 GB, 80 GB
**Q15. How long can you use these Data Center GPUs for?** On an individual or per-project basis. Select 1 option per row.
Answers (Rows): 1 GPU, 2 GPUs, 4 GPUs, 8 GPUs, 16 GPUs, 32 GPUs, 64+ GPUs
Answers (Columns): N/A, Hours, Days, Weeks, Months, Indefinitely

---

**Q16. Does your system have any GPU-to-GPU connections?**
Answers: No, Not sure, NVLink, NVSwitch, AMD Infinity Fabric
**Q17. What kind of (inter-machine) networking connections do you have?**
Answers: Not sure, No multi-node support, Multi-node supported (but connectivity is unknown), Ethernet (throughput unknown), Ethernet (1 Gbps), Ethernet (10 Gbps), Ethernet (25 Gbps), Ethernet (40/50 Gbps), Ethernet (100 Gbps), Ethernet (200 Gbps), Ethernet (400 Gbps), Infiniband (throughput unknown), Infiniband QDR (32 Gbps), Infiniband FDR (54 Gbps), Infiniband EDR (100 Gbps), Infiniband HDR (200 Gbps), Infiniband NDR (400 Gbps)
**Q18. What is the largest model you have pre-trained from scratch (if any)?**
**Q19. Would you like to provide any other details regarding your current access to compute?**

---

Table 2: Questions in our academic compute survey. "Answer" entails single choice and "Answers" entails multiple choices. Remaining questions (and "[Other]" choice) are fill-in-the-blank.

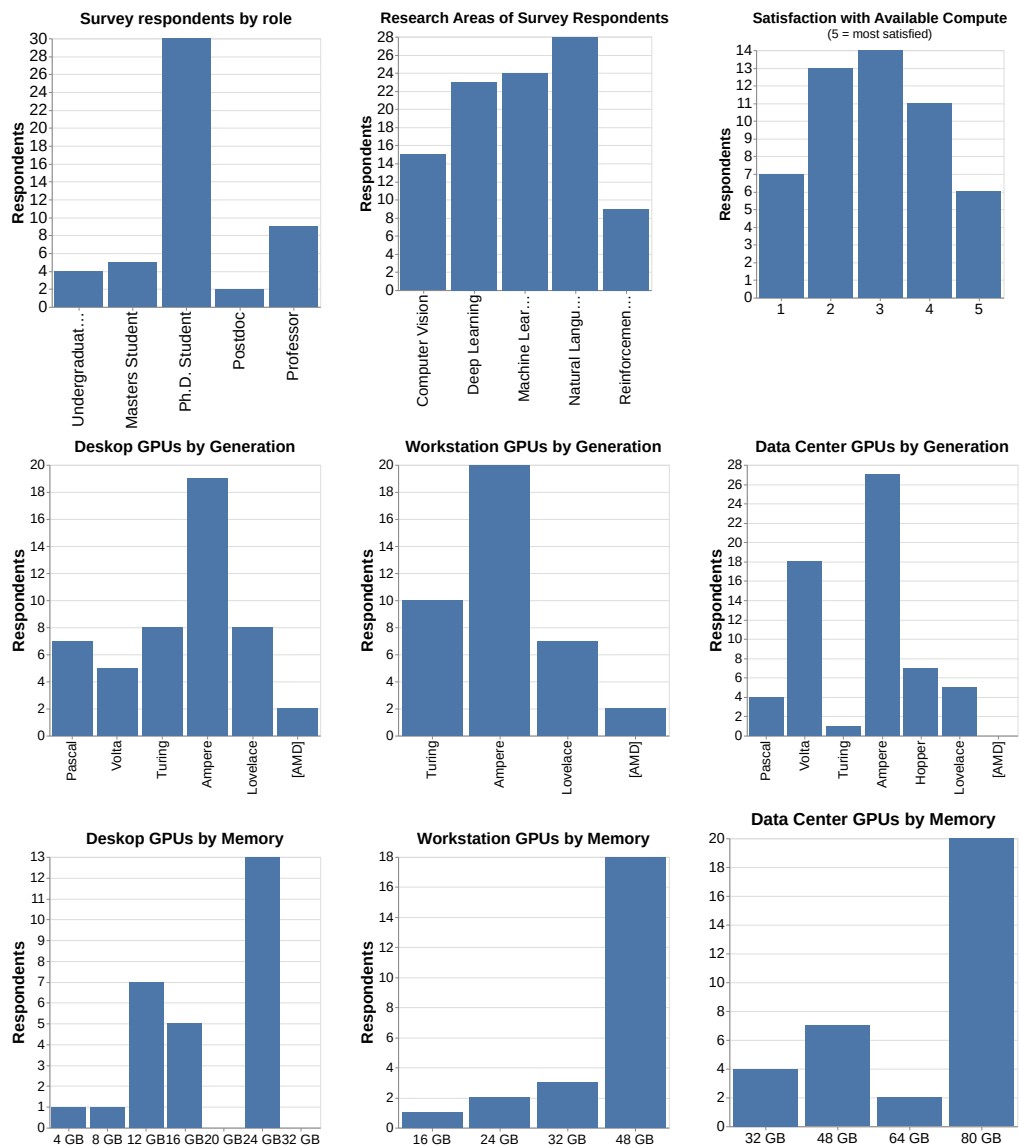

Figure 7: Results from our survey. We also find that 70% of respondents have inter-GPU connections (e.g. NVLink). 33% of respondents reported having Ethernet and 24% reported having Infiniband inter-node connections.

## B  Software and Hardware Specifications

Our codebase is written in Python and primarily uses PyTorch (Paszke et al., 2019) and Hugging Face Transformers (Wolf et al., 2020). In each of our benchmarks, we allocate 4 CPU cores and 64 GB of system memory for each allocated GPU. We assume that training data will be pre-processed and cached ahead-of-time, ensuring that the training process is GPU-bound (i.e. negligibly bound by data loading from disk, rather than by on-the-fly data pre-processing on CPUs).

Our cluster is managed with SLURM. All nodes have 8 GPUs and are connected via Infiniband (200–400 Gb/s).

1. Each node with 8 x 24GB RTX 3090 GPUs has 64 CPUs with 1TB system memory.

2. Each node with 8 x 48GB A6000 GPUs has 64 CPUs with 1TB system memory. Pairs of GPUs are connected with NVLink.

3. Each node with 8 x 80GB A100 GPUs (SXM) has 128 CPUs with 1TB system memory. GPUs are fully-connected with NVSwitch.

4. Each node with 8 x 80GB H100 GPUs (SXM) has 112 CPUs with 2TB system memory. GPUs are fully-connected with NVSwitch.

## C  Model Hyper-parameters for Replication

| Model | Params. | Seq. Len. / Vocab. Size | Image Size / Classes | Batch Size | Training Steps | Mixed Precision | Optimizer |
|---|---|---|---|---|---|---|---|
| Pythia | 160M | 2049 / 50K | — | 1024 | 143K | fp16 | Adam |
| | 410M | | — | | | fp16 | |
| | 1.0B | | — | | | bf16 | |
| | 2.8B | | — | | | fp16 | |
| | 6.9B | | — | | | fp16 | |
| RoBERTa | 360M | 512 / 50K | — | 8192 | 500K | fp16 | Adam |
| Mamba | 2.8B | 4096 / 50K | — | 128 | 572K | bf16 | AdamW |
| ConvNeXt | 390M | — | 224 / 22K | 4096 | 312K | N/A | AdamW |
| ViT | 330M | 256 / — | 224 / 22K | 4096 | 312K | N/A | Adam |

Table 3: Model & training hyper-parameters from original reports. We use and list further hyper-parameters (e.g. learning rate & scheduler)—which are necessary for replication, but don't affect our measurements—in our codebase.

## D  Additional Design Decisions

**Choice of software.**  We run experiments in Python and PyTorch on GPUs (the most prevalent compute stack in the academic AI research community). Rather than writing a custom training loop, we rely on a training API (i.e. the Trainer from Transformers, which already supports many features and receives active maintenance) to inherit future upstream implementations—like models and training optimizations—with minimal additional implementation in our code. We choose the Transformers library [over alternatives, like MosaicML Composer (The Mosaic ML Team, 2021) and PyTorch Lightning (Falcon and the PyTorch Lightning team, 2019)], because it has native support for a very large library of models and (at this time) has the most complete support for the efficient training methods in our investigation.

**Other efficient training methods.**  Mixed precision training (Micikevicius et al., 2018) improves speed—as GPUs compute 16-bit operations more quickly—with variable effects on memory consumption. This method is rather common and we use the original model's mixed precision setting for replication purposes. Alternative optimizers reduce memory (Dettmers et al., 2021; Li et al., 2023; Shazeer & Stern, 2018; Luo et al., 2023) or communication volume (Tang et al., 2021; Li et al., 2021). Low-rank (Lialin et al., 2023), quantized (Micikevicius et al., 2022), and sparse (PyTorch, 2024) training can both reduce memory and improve speed.

Tensor (Shazeer et al., 2018; Shoeybi et al., 2019), pipeline (Huang et al., 2018; Narayanan et al., 2019), and 3D (Narayanan et al., 2021) parallelism are alternatives to the model sharding described in our paper, but require extensive, model-specific implementations. Dataset distillation (Wang et al., 2018) and model weight averaging (Kaddour, 2022) can improve model convergence rates. Finally, Kaddour et al. (2023) more closely tests several methods (not listed above) and finds that their performance gains vanish under controlled compute budgets.

We exclude all such methods from our approach, as they may either change a model's optimization formula or require extensive model-specific implementations.

**Other models.**  We don't consider models with more than 7B parameters, because we find that Pythia-6.9B already requires 30 days to train on the very best "academic" hardware (8

H100 GPUs). Finally, other models may be similar (e.g. in size and architecture) to those we report, but may be trained on much larger quantities of data, directly inflating the training time. For example, OLMo-1B (Groeneveld et al., 2024) is trained on 10 times as many tokens as Pythia-1B.

# E    Model Compute & GPU Throughput

| Model | Training FLOPs |
|---|---|
| Mamba | $8.7 \times 10^{20}$ |
| RoBERTa | $4.8 \times 10^{21}$ |
| Pythia (160M) | $2.9 \times 10^{20}$ |
| Pythia (410M) | $8.2 \times 10^{20}$ |
| Pythia (1B) | $1.9 \times 10^{21}$ |
| Pythia (2.8B) | $5.4 \times 10^{21}$ |
| Pythia (6.9B) | $1.3 \times 10^{22}$ |
| ConvNeXt | $1.4 \times 10^{21}$ |
| ViT | $4.7 \times 10^{20}$ |

(a)

| GPU | FLOPs/sec |
|---|---|
| RTX 3090 | $7.1 \times 10^{13}$ |
| A6000 | $1.6 \times 10^{14}$ |
| A100 | $3.1 \times 10^{14}$ |
| H100 | $7.6 \times 10^{14}$ |

(b)

Figure 8: (a) Total training compute for models in our investigation. (b) 16-bit throughput specifications for GPUs in our investigation. Should be multiplied by the total number of GPUs for total throughput.

# F    Optimization Methods and Search Space

Our memory-saving methods have several options:

1. Activation Checkpointing [True / False]
2. Model Sharding [stages 0–3 with Zero / FSDP]
3. Offloading [True / False]

Regarding sharding: Stage (0) indicates sharding is disabled. Stage (1) shards only optimizer states, (2) shards optimizer states and gradients, (3) shards optimizer states, gradients, and parameter weights. Deepspeed offers the Zero implementation (stages 1–3) and Pytorch offers the FSDP implementation (stages 2–3).

These methods have additional constraints:

- Offloading is only allowed when sharding is enabled.
- Model sharding is a no-op when using 1 GPU, so we exclude configurations with sharding (and no offloading) in that setting.
- Only optimizer states are offloaded if the sharding stage is (1) or (2); both optimizer states and parameter weights are offloaded if the stage is (3).

Altogether, these methods form a search space of 12 options (when using 1 GPU) and 22 options (when using >1 GPU). Furthermore:

- TF32 mode is only available on Ampere and newer generation GPUs.
- At the time of experiments, the Mamba model does not support torch.compile and Roberta does not have an available implementation for the Flash Attention custom kernel. torch.compile is not compatible with Deepspeed and so we do not enable model compilation when sharding with Zero.

- Compilation only improves speed and does not reduce GPU memory consumption. It incurs an up-front time cost (usually a few minutes) and re-compilation is necessary for every tested per-gpu batch size. To save significant time in our benchmarks, we do not compile the model when finding the maximum batch size — where compilation is irrelevant.

# G   Naive Training Times & Deaggregated Results

| | | RTX 3090 (24 GB) | | | | A6000 (48 GB) | | | | A100 (80 GB) | | | | H100 (80 GB) | | | |
| --- | --- | --- | --- | --- | --- | --- | --- | --- | --- | --- | --- | --- | --- | --- | --- | --- | --- |
| | | 1 | 2 | 4 | 8 | 1 | 2 | 4 | 8 | 1 | 2 | 4 | 8 | 1 | 2 | 4 | 8 |
| Pythia | 160M | 125 | 92 | 46 | 20 | 124 | 63 | 35 | 19 | 71 | 36 | 18 | 9 | 42 | 21 | 11 | 5 |
| | 410M | 430 | 431 | 226 | 69 | 421 | 213 | 150 | 64 | 232 | 117 | 59 | 30 | 137 | 69 | 35 | 18 |
| | 1B | — | — | — | — | 326 | 238 | 82 | 41 | 160 | 81 | 41 | 21 | 90 | 45 | 23 | 12 |
| | 2.8B | — | — | — | — | — | — | — | — | — | — | — | — | — | — | — | — |
| | 6.9B | — | — | — | — | — | — | — | — | — | — | — | — | — | — | — | — |
| RoBERTa | 350M | 1560 | 1095 | 418 | 420 | 1375 | 1125 | 489 | 185 | 685 | 345 | 174 | 87 | 399 | 201 | 101 | 51 |
| Mamba | 2.8B | — | — | — | — | — | — | — | — | — | — | — | — | — | — | — | — |
| ConvNeXt | 390M | 240 | 196 | 60 | 30 | 233 | 166 | 68 | 30 | 236 | 119 | 60 | 31 | — | 49 | 25 | 13 |
| ViT | 325M | 255 | 218 | 63 | 33 | 243 | 158 | 72 | 31 | 246 | 124 | 62 | 32 | 93 | 47 | 24 | 12 |

Table 4: Training times (in days) for model–GPU combinations using naive settings. "—" indicates infeasibility with all efficient methods.

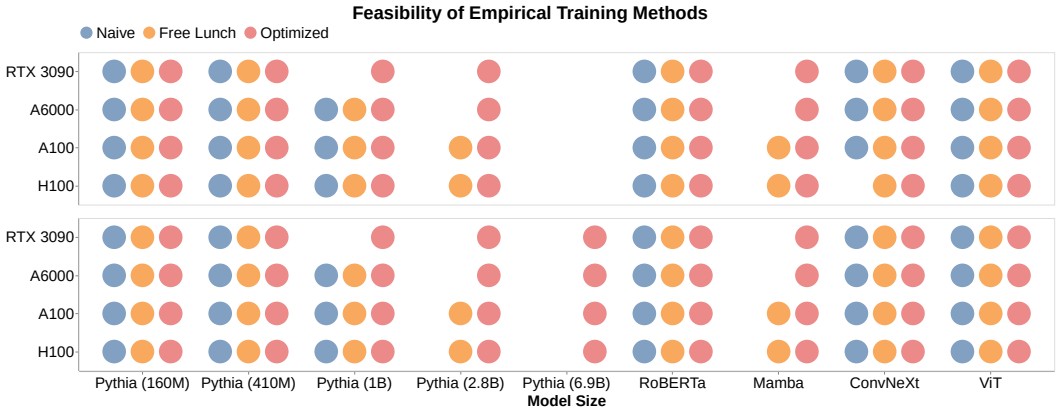

Figure 9: Extended visualization of feasible empirical training methods (showing all models, as well as free-lunch only methods). We show 1 GPU setting (top) and the 2+ GPU setting (bottom). For certain highly-constrained settings (e.g. Pythia-6.9B), we observe a gap between the 1 and 2+ GPU setting, where all combinations of memory-saving methods are infeasible for the former. As in Fig. 2b, there exists a feasible combination of memory-saving methods for all models and we see a larger need for these methods with larger models and smaller GPUs.

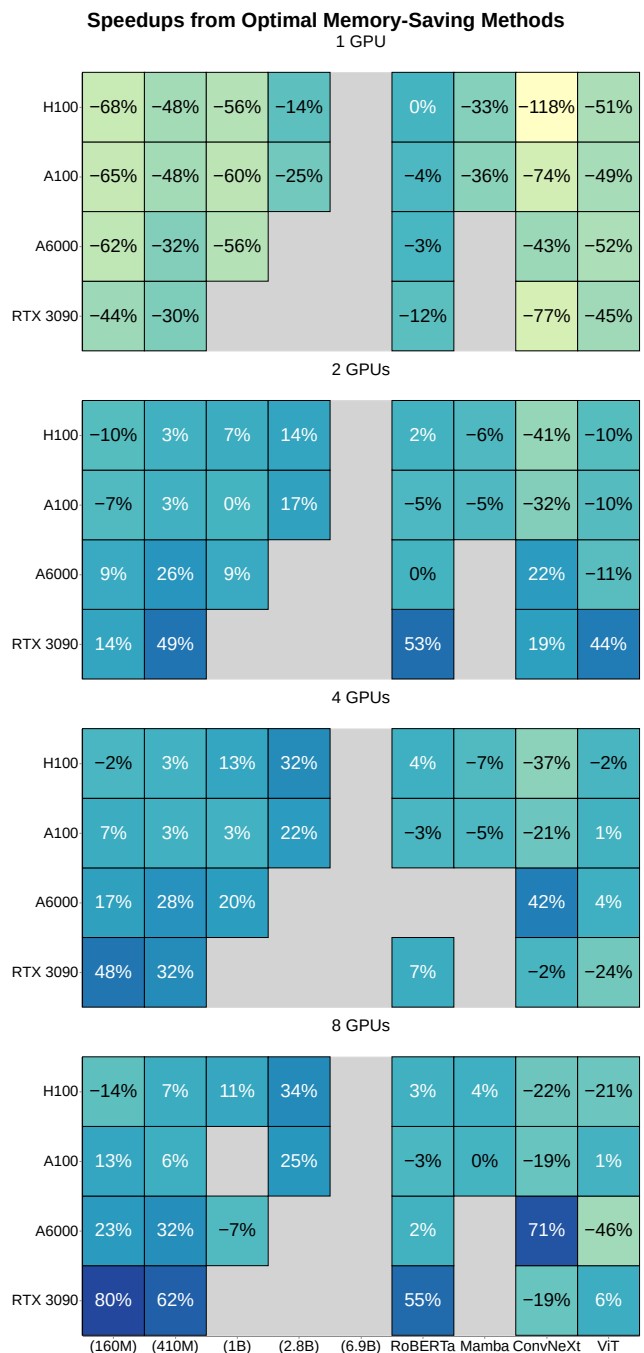

Figure 10: Training time speedups (or slowdowns if negative) for optimal memory-saving methods over using no memory-saving methods (i.e. only free-lunch). Results are shown for our full set of models and deaggregated by number of GPUs. Parenthesized sizes refer to Pythia models. Gray indicates infeasibility when using free-lunch methods alone (and sometimes with all memory-saving methods). As in Fig. 3, we see consistent degradations when using these methods in the 1 GPU setting (where model sharding is a no-op). Here, we can also see that these methods don't benefit the particular computer vision models selected in our investigation as strongly. These models simply require less memory than Pythia (even at similar parameter sizes, due to other differences in hyper-parameters e.g. in sequence length and batch size). Thus, they see fewer benefits from memory-saving methods.

## H   Current Hardware Costs

We show quotes for different GPUs and machine builds (supporting different numbers of GPUs) here. We provide reasonable estimates for general builds and leave room for exploitation (e.g. finding more optimal prices for more specific sets of components). Our findings are a guide: we recommend end-users obtain more recent quotes (and benchmark their own models) before making significant investments in hardware.

|         | Cost     | Quote |
|---------|----------|-------|
| RTX3090 | $1,300   | Resale sites (e.g. eBay) |
| A6000   | $4,800   | Online retailers (e.g. Newegg) |
| A100    | $19,000  | https://www.thinkmate.com/systems/servers/gpx |
| H100    | $30,000  | https://www.thinkmate.com/systems/servers/gpx |

Table 5: Costs per GPU (quoted on September 21, 2024).

|       | Part         | Detail                         | Cost       |
|-------|--------------|--------------------------------|------------|
| 1 GPU | CPU          | Intel Core i3-10300            | $99.00     |
|       | CPU Cooler   | Noctua NH-D15                  | $109.95    |
|       | Motherboard  | MSI Z490-A PRO                 | $329.00    |
|       | Memory       | Corsair Vengeance LPX 64 GB    | $102.49    |
|       | Storage      | Western Digital - 4 TB         | $139.99    |
|       | Case         | NZXT H7 Flow (2022)            | $99.99     |
|       | Power Supply | Corsair RM1000x (2021)         | $139.99    |
|       |              | **Total**                      | $1,020.41  |
| 2 GPU | CPU          | Intel Core i7-11700K           | $247.93    |
|       | CPU Cooler   | Noctua NH-D15                  | $109.95    |
|       | Motherboard  | MSI Z490-A PRO                 | $329.00    |
|       | Memory       | Corsair Vengeance LPX 128 GB   | $252.52    |
|       | Storage      | Western Digital - 4 TB         | $139.99    |
|       | Case         | NZXT H7 Flow (2022)            | $99.99     |
|       | Power Supply | EVGA SuperNOVA 1600 G+         | $276.91    |
|       |              | **Total**                      | $1,456.29  |

Table 6: Desktop costs (compatible with 1 or 2 GPUs). Quotes from consumer retailers, as of September 21, 2024.

| Server | Part         | Detail                               | Cost        |
|--------|--------------|--------------------------------------|-------------|
| 4 GPU  | Barebone     | 1U GPU Server (Intel C621A Chipset)  | $7,482.00   |
|        | Processor    | 2 x Intel Xeon Silver 4310 (12-Core) |             |
|        | Memory       | 16 x 16GB DDR4                       |             |
|        | Power Supply | 2 x 2200W 80 Plus Titanium           |             |
|        | Hard Drive   | Western Digital - 4 TB               |             |
| 8 GPU  | BareBone     | 2U GPU Server (Intel C621A Chipset)  | $10,673.00  |
|        | Processor    | 2 x Intel Xeon Silver 4314 (16-Core) |             |
|        | Memory       | 16 x 32GB DDR4                       |             |
|        | Power Supply | 2 x 3200W 80 Plus Platinum           |             |
|        | Hard Drive   | Western Digital - 4 TB               |             |

Table 7: Server costs (compatible with 4 or 8 GPUs). Quotes from https://www.thinkmate.com/system/gpx-xn4-21s3-4gpu and https://www.thinkmate.com/system/gpx-xh8-22s3-8gpu on September 21, 2024.

# I  Extended Cost–Benefit Results

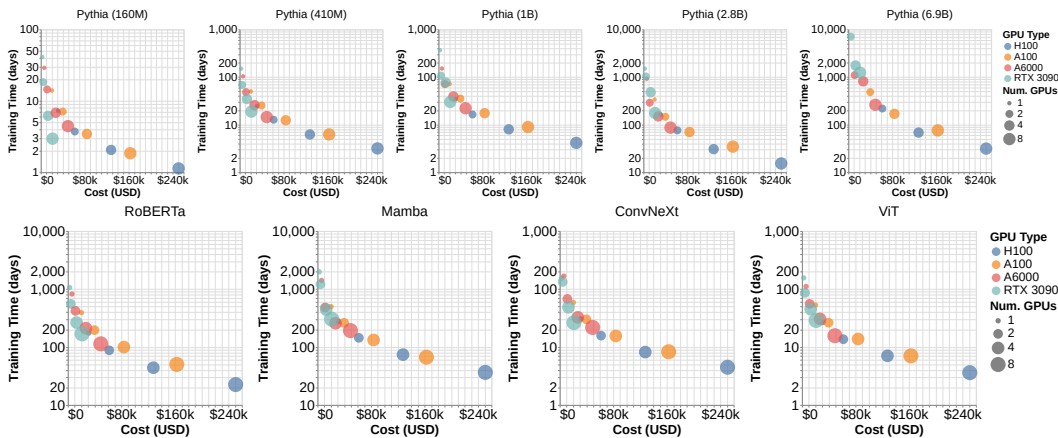

Figure 11: Comparing hardware by overall cost and training time, for all models. Y-axis is log-scale to clearly capture outliers.

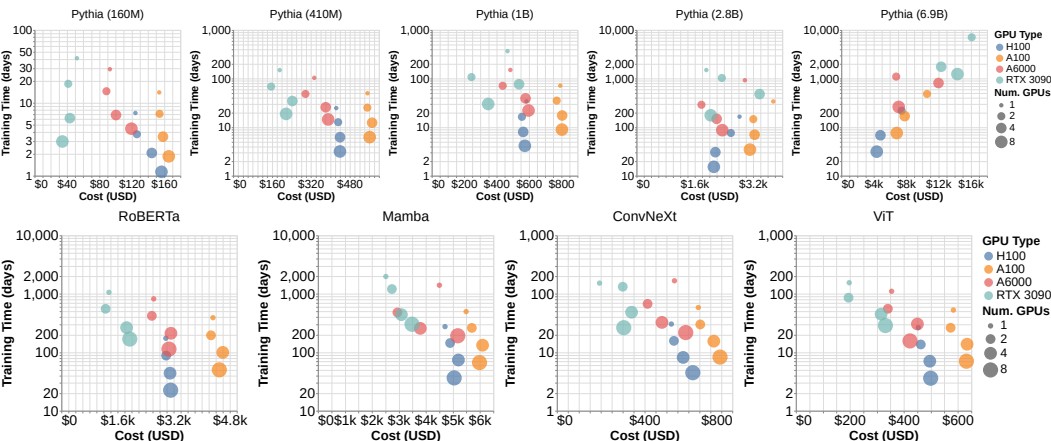

Figure 12: Comparing hardware by experiment-normalized cost and training time, for all models. Y-axis is log scale to clearly capture outliers.

# J  Optimal Training Settings

In this table, "MBS" refers to the batch size per GPU and "GAS" refers to the number of gradient accumulation steps. Our effective batch size is # GPUs × MBS × GAS. This matches our desired value for replication (Appendix C). Settings neglected from this table are infeasible using all methods.

| Model | GPU Type | # GPUs | Free Lunch | Checkpointing | Sharding | Offloading | MBS | GAS |
|---|---|---|---|---|---|---|---|---|
| Pythia (160M) | RTX 3090 | 1 | ✓ | ✗ | — | ✗ | 8 | 128 |
| Pythia (160M) | RTX 3090 | 2 | ✓ | ✗ | Zero (1) | ✗ | 8 | 64 |
| Pythia (160M) | RTX 3090 | 4 | ✓ | ✗ | Zero (1) | ✗ | 8 | 32 |
| Pythia (160M) | RTX 3090 | 8 | ✓ | ✗ | Zero (1) | ✗ | 8 | 16 |
| Pythia (160M) | A6000 | 1 | ✓ | ✗ | — | ✗ | 16 | 64 |
| Pythia (160M) | A6000 | 2 | ✓ | ✗ | Zero (1) | ✗ | 16 | 32 |
| Pythia (160M) | A6000 | 4 | ✓ | ✗ | Zero (1) | ✗ | 32 | 8 |

| Model | GPU Type | # GPUs | Free Lunch | Checkpointing | Sharding | Offloading | MBS | GAS |
|---|---|---|---|---|---|---|---|---|
| Pythia (160M) | A6000 | 8 | ✓ | ✗ | Zero (1) | ✗ | 32 | 4 |
| Pythia (160M) | A100 | 1 | ✓ | ✗ | — | ✗ | 32 | 32 |
| Pythia (160M) | A100 | 2 | ✓ | ✗ | — | ✗ | 32 | 16 |
| Pythia (160M) | A100 | 4 | ✓ | ✗ | Zero (1) | ✗ | 32 | 8 |
| Pythia (160M) | A100 | 8 | ✓ | ✗ | Zero (1) | ✗ | 32 | 4 |
| Pythia (160M) | H100 | 1 | ✓ | ✗ | — | ✗ | 32 | 32 |
| Pythia (160M) | H100 | 2 | ✓ | ✗ | — | ✗ | 32 | 16 |
| Pythia (160M) | H100 | 4 | ✓ | ✗ | — | ✗ | 32 | 8 |
| Pythia (160M) | H100 | 8 | ✓ | ✗ | — | ✗ | 32 | 4 |
| Pythia (410M) | RTX 3090 | 1 | ✓ | ✗ | — | ✗ | 4 | 256 |
| Pythia (410M) | RTX 3090 | 2 | ✓ | ✗ | Zero (1) | ✗ | 4 | 128 |
| Pythia (410M) | RTX 3090 | 4 | ✓ | ✗ | Zero (1) | ✗ | 4 | 64 |
| Pythia (410M) | RTX 3090 | 8 | ✓ | ✗ | Zero (1) | ✗ | 8 | 16 |
| Pythia (410M) | A6000 | 1 | ✓ | ✗ | — | ✗ | 8 | 128 |
| Pythia (410M) | A6000 | 2 | ✓ | ✗ | Zero (1) | ✗ | 16 | 32 |
| Pythia (410M) | A6000 | 4 | ✓ | ✗ | Zero (1) | ✗ | 16 | 16 |
| Pythia (410M) | A6000 | 8 | ✓ | ✗ | Zero (1) | ✗ | 16 | 8 |
| Pythia (410M) | A100 | 1 | ✓ | ✗ | — | ✗ | 16 | 64 |
| Pythia (410M) | A100 | 2 | ✓ | ✗ | Zero (1) | ✗ | 16 | 32 |
| Pythia (410M) | A100 | 4 | ✓ | ✗ | Zero (1) | ✗ | 16 | 16 |
| Pythia (410M) | A100 | 8 | ✓ | ✗ | Zero (1) | ✗ | 16 | 8 |
| Pythia (410M) | H100 | 1 | ✓ | ✗ | — | ✗ | 16 | 64 |
| Pythia (410M) | H100 | 2 | ✓ | ✗ | Zero (1) | ✗ | 16 | 32 |
| Pythia (410M) | H100 | 4 | ✓ | ✗ | Zero (1) | ✗ | 16 | 16 |
| Pythia (410M) | H100 | 8 | ✓ | ✗ | Zero (1) | ✗ | 16 | 8 |
| Pythia (1B) | RTX 3090 | 1 | ✓ | ✓ | — | ✗ | 2 | 512 |
| Pythia (1B) | RTX 3090 | 2 | ✓ | ✗ | Zero (1) | ✗ | 4 | 128 |
| Pythia (1B) | RTX 3090 | 4 | ✓ | ✓ | Zero (1) | ✗ | 16 | 16 |
| Pythia (1B) | RTX 3090 | 8 | ✓ | ✗ | Zero (1) | ✗ | 4 | 32 |
| Pythia (1B) | A6000 | 1 | ✓ | ✗ | — | ✗ | 4 | 256 |
| Pythia (1B) | A6000 | 2 | ✓ | ✗ | Zero (1) | ✗ | 8 | 64 |
| Pythia (1B) | A6000 | 4 | ✓ | ✗ | Zero (1) | ✗ | 8 | 32 |
| Pythia (1B) | A6000 | 8 | ✓ | ✗ | — | ✗ | 4 | 32 |
| Pythia (1B) | A100 | 1 | ✓ | ✗ | — | ✗ | 8 | 128 |
| Pythia (1B) | A100 | 2 | ✓ | ✗ | Zero (1) | ✗ | 16 | 32 |
| Pythia (1B) | A100 | 4 | ✓ | ✗ | Zero (1) | ✗ | 16 | 16 |
| Pythia (1B) | A100 | 8 | ✓ | ✗ | Zero (1) | ✗ | 16 | 8 |
| Pythia (1B) | H100 | 1 | ✓ | ✗ | — | ✗ | 8 | 128 |
| Pythia (1B) | H100 | 2 | ✓ | ✗ | Zero (1) | ✗ | 16 | 32 |
| Pythia (1B) | H100 | 4 | ✓ | ✗ | Zero (1) | ✗ | 16 | 16 |
| Pythia (1B) | H100 | 8 | ✓ | ✗ | Zero (1) | ✗ | 16 | 8 |
| Pythia (2.8B) | RTX 3090 | 1 | ✓ | ✓ | Zero (1) | ✓ | 8 | 128 |
| Pythia (2.8B) | RTX 3090 | 2 | ✓ | ✓ | Zero (3) | ✓ | 16 | 32 |
| Pythia (2.8B) | RTX 3090 | 4 | ✓ | ✓ | Zero (3) | ✓ | 16 | 16 |
| Pythia (2.8B) | RTX 3090 | 8 | ✓ | ✗ | Zero (1) | ✗ | 1 | 128 |
| Pythia (2.8B) | A6000 | 1 | ✓ | ✓ | Zero (1) | ✓ | 32 | 32 |
| Pythia (2.8B) | A6000 | 2 | ✓ | ✗ | Zero (1) | ✗ | 2 | 256 |
| Pythia (2.8B) | A6000 | 4 | ✓ | ✗ | Zero (1) | ✗ | 4 | 64 |
| Pythia (2.8B) | A6000 | 8 | ✓ | ✗ | Zero (1) | ✗ | 4 | 32 |
| Pythia (2.8B) | A100 | 1 | ✓ | ✗ | — | ✗ | 2 | 512 |
| Pythia (2.8B) | A100 | 2 | ✓ | ✗ | Zero (1) | ✗ | 8 | 64 |
| Pythia (2.8B) | A100 | 4 | ✓ | ✗ | Zero (1) | ✗ | 8 | 32 |
| Pythia (2.8B) | A100 | 8 | ✓ | ✗ | Zero (1) | ✗ | 8 | 16 |
| Pythia (2.8B) | H100 | 1 | ✓ | ✗ | — | ✗ | 2 | 512 |
| Pythia (2.8B) | H100 | 2 | ✓ | ✗ | Zero (1) | ✗ | 8 | 64 |
| Pythia (2.8B) | H100 | 4 | ✓ | ✗ | Zero (1) | ✗ | 8 | 32 |
| Pythia (2.8B) | H100 | 8 | ✓ | ✗ | Zero (1) | ✗ | 8 | 16 |
| Pythia (6.9B) | RTX 3090 | 2 | ✓ | ✗ | FSDP (3) | ✓ | 1 | 512 |
| Pythia (6.9B) | RTX 3090 | 4 | ✓ | ✓ | Zero (3) | ✓ | 8 | 32 |
| Pythia (6.9B) | RTX 3090 | 8 | ✓ | ✓ | Zero (3) | ✗ | 4 | 32 |
| Pythia (6.9B) | A6000 | 2 | ✓ | ✓ | Zero (1) | ✓ | 16 | 32 |
| Pythia (6.9B) | A6000 | 4 | ✓ | ✓ | Zero (3) | ✓ | 32 | 8 |
| Pythia (6.9B) | A6000 | 8 | ✓ | ✓ | Zero (1) | ✗ | 16 | 8 |
| Pythia (6.9B) | A100 | 2 | ✓ | ✓ | Zero (2) | ✓ | 32 | 16 |
| Pythia (6.9B) | A100 | 4 | ✓ | ✗ | Zero (1) | ✗ | 4 | 64 |

| Model | GPU Type | # GPUs | Free Lunch | Checkpointing | Sharding | Offloading | MBS | GAS |
|-------|----------|--------|------------|---------------|----------|------------|-----|-----|
| Pythia (6.9B) | A100 | 8 | ✓ | ✗ | Zero (1) | ✗ | 4 | 32 |
| Pythia (6.9B) | H100 | 2 | ✓ | ✓ | FSDP (2) | ✗ | 4 | 128 |
| Pythia (6.9B) | H100 | 4 | ✓ | ✗ | Zero (1) | ✗ | 4 | 64 |
| Pythia (6.9B) | H100 | 8 | ✓ | ✗ | Zero (1) | ✗ | 4 | 32 |
| RoBERTa | RTX 3090 | 1 | ✓ | ✗ | — | ✗ | 8 | 1024 |
| RoBERTa | RTX 3090 | 2 | ✓ | ✗ | Zero (1) | ✗ | 16 | 256 |
| RoBERTa | RTX 3090 | 4 | ✓ | ✗ | Zero (1) | ✗ | 16 | 128 |
| RoBERTa | RTX 3090 | 8 | ✓ | ✓ | — | ✗ | 32 | 32 |
| RoBERTa | A6000 | 1 | ✓ | ✗ | — | ✗ | 16 | 512 |
| RoBERTa | A6000 | 2 | ✓ | ✗ | — | ✗ | 16 | 256 |
| RoBERTa | A6000 | 4 | ✓ | ✓ | — | ✗ | 64 | 32 |
| RoBERTa | A6000 | 8 | ✓ | ✗ | Zero (1) | ✗ | 32 | 32 |
| RoBERTa | A100 | 1 | ✓ | ✗ | — | ✗ | 32 | 256 |
| RoBERTa | A100 | 2 | ✓ | ✗ | — | ✗ | 32 | 128 |
| RoBERTa | A100 | 4 | ✓ | ✗ | — | ✗ | 32 | 64 |
| RoBERTa | A100 | 8 | ✓ | ✗ | — | ✗ | 32 | 32 |
| RoBERTa | H100 | 1 | ✓ | ✓ | — | ✗ | 128 | 64 |
| RoBERTa | H100 | 2 | ✓ | ✓ | — | ✗ | 128 | 32 |
| RoBERTa | H100 | 4 | ✓ | ✓ | — | ✗ | 128 | 16 |
| RoBERTa | H100 | 8 | ✓ | ✓ | — | ✗ | 128 | 8 |
| Mamba | RTX 3090 | 1 | ✓ | ✓ | FSDP (2) | ✓ | 4 | 32 |
| Mamba | RTX 3090 | 2 | ✓ | ✓ | FSDP (3) | ✓ | 4 | 16 |
| Mamba | RTX 3090 | 4 | ✓ | ✓ | Zero (1) | ✗ | 1 | 32 |
| Mamba | RTX 3090 | 8 | ✓ | ✓ | FSDP (2) | ✗ | 2 | 8 |
| Mamba | A6000 | 1 | ✓ | ✗ | Zero (2) | ✓ | 2 | 64 |
| Mamba | A6000 | 2 | ✓ | ✗ | Zero (1) | ✗ | 1 | 64 |
| Mamba | A6000 | 4 | ✓ | ✗ | Zero (1) | ✗ | 2 | 16 |
| Mamba | A6000 | 8 | ✓ | ✗ | Zero (1) | ✗ | 2 | 8 |
| Mamba | A100 | 1 | ✓ | ✗ | — | ✗ | 1 | 128 |
| Mamba | A100 | 2 | ✓ | ✗ | — | ✗ | 1 | 64 |
| Mamba | A100 | 4 | ✓ | ✗ | — | ✗ | 1 | 32 |
| Mamba | A100 | 8 | ✓ | ✗ | — | ✗ | 1 | 16 |
| Mamba | H100 | 1 | ✓ | ✗ | — | ✗ | 1 | 128 |
| Mamba | H100 | 2 | ✓ | ✗ | — | ✗ | 1 | 64 |
| Mamba | H100 | 4 | ✓ | ✗ | — | ✗ | 1 | 32 |
| Mamba | H100 | 8 | ✓ | ✗ | Zero (1) | ✗ | 4 | 4 |
| ConvNeXt | RTX 3090 | 1 | ✓ | ✗ | — | ✗ | 32 | 128 |
| ConvNeXt | RTX 3090 | 2 | ✓ | ✗ | Zero (1) | ✗ | 32 | 64 |
| ConvNeXt | RTX 3090 | 4 | ✓ | ✗ | — | ✗ | 32 | 32 |
| ConvNeXt | RTX 3090 | 8 | ✓ | ✗ | — | ✗ | 32 | 16 |
| ConvNeXt | A6000 | 1 | ✓ | ✗ | — | ✗ | 64 | 64 |
| ConvNeXt | A6000 | 2 | ✓ | ✗ | Zero (1) | ✗ | 64 | 32 |
| ConvNeXt | A6000 | 4 | ✓ | ✗ | Zero (1) | ✗ | 64 | 16 |
| ConvNeXt | A6000 | 8 | ✓ | ✓ | — | ✗ | 128 | 4 |
| ConvNeXt | A100 | 1 | ✓ | ✗ | — | ✗ | 128 | 32 |
| ConvNeXt | A100 | 2 | ✓ | ✗ | — | ✗ | 128 | 16 |
| ConvNeXt | A100 | 4 | ✓ | ✗ | — | ✗ | 128 | 8 |
| ConvNeXt | A100 | 8 | ✓ | ✗ | — | ✗ | 128 | 4 |
| ConvNeXt | H100 | 1 | ✓ | ✗ | — | ✗ | 128 | 32 |
| ConvNeXt | H100 | 2 | ✓ | ✗ | — | ✗ | 128 | 16 |
| ConvNeXt | H100 | 4 | ✓ | ✗ | — | ✗ | 128 | 8 |
| ConvNeXt | H100 | 8 | ✓ | ✗ | — | ✗ | 128 | 4 |
| ViT | RTX 3090 | 1 | ✓ | ✗ | — | ✗ | 32 | 128 |
| ViT | RTX 3090 | 2 | ✓ | ✗ | Zero (1) | ✗ | 32 | 64 |
| ViT | RTX 3090 | 4 | ✓ | ✗ | — | ✗ | 32 | 32 |
| ViT | RTX 3090 | 8 | ✓ | ✗ | Zero (1) | ✗ | 32 | 16 |
| ViT | A6000 | 1 | ✓ | ✗ | — | ✗ | 128 | 32 |
| ViT | A6000 | 2 | ✓ | ✗ | — | ✗ | 128 | 16 |
| ViT | A6000 | 4 | ✓ | ✗ | Zero (1) | ✗ | 128 | 8 |
| ViT | A6000 | 8 | ✓ | ✗ | — | ✗ | 128 | 4 |
| ViT | A100 | 1 | ✓ | ✗ | — | ✗ | 128 | 32 |
| ViT | A100 | 2 | ✓ | ✗ | — | ✗ | 128 | 16 |
| ViT | A100 | 4 | ✓ | ✗ | Zero (1) | ✗ | 128 | 8 |
| ViT | A100 | 8 | ✓ | ✗ | Zero (1) | ✗ | 128 | 4 |
| ViT | H100 | 1 | ✓ | ✗ | — | ✗ | 128 | 32 |
| ViT | H100 | 2 | ✓ | ✗ | — | ✗ | 128 | 16 |

| Model | GPU Type | # GPUs | Free Lunch | Checkpointing | Sharding | Offloading | MBS | GAS |
|---|---|---|---|---|---|---|---|---|
| ViT | H100 | 4 | ✓ | ✗ | — | ✗ | 128 | 8 |
| ViT | H100 | 8 | ✓ | ✗ | — | ✗ | 128 | 4 |

