# OpenReview forum: "$100K or 100 Days: Trade-offs when Pre-Training with Academic Resources"
_colmweb.org/COLM/2025/Conference — COLM 2025_

### Official Review · Reviewer_QD5y · 2025-05-08

**Rating:** 9
**Confidence:** 4
**Ethics Flag:** 1

**Summary:**

The paper conducted a comprehensive empirical study on the pre-training with academic resources. It firstly surveys the academic people to learn about their compute resources. Then, under the typical academic settings of the resources, it replicates the pre-training of many models and also introduces several acceleration methods. It also concludes a cost-benefit analysis to provides insights on the price and pretraining time tradeoff.

**Reasons To Accept:**

1. The paper studies a very important and very practical problem, i.e., the pretraining with academic resources.
2. The experimental results of this paper will shed light on the practice of the academic pretraining and reveal a brighter picture to this field.
3. The acceleration methods introduced will help academic researchers boost the training efficiency.
4. The paper is well written and very easy to follow.

**Reasons To Reject:**

1. The paper lacks original innovation and does not propose any novel techniques.

---

> ### Author Response · Authors · 2025-05-30
>
> Dear [QD5y], thank you very much for your review. We appreciate your shared belief about the importance of our research problem. We comment further on the novelty of our work in [our general comment](https://openreview.net/forum?id=EFxC34XbDh&noteId=xTpguwdEAr).

---

> > ### Comment · Reviewer_QD5y · 2025-06-10
> >
> > Thank you for your response. I would like to maintain my score.

---

### Official Review · Reviewer_DTxJ · 2025-05-11

**Rating:** 6
**Confidence:** 4
**Ethics Flag:** 1

**Summary:**

This work investigates whether academic labs with limited compute can feasibly pre-train large models. Through surveys, benchmarks, and empirical optimizations, the authors provide insights into training efficiency trade-offs and show that even with modest academic hardware, substantial pre-training is possible. Code is provided as supp materials to help the community adopt and learn mentioned techniques. Multiple engineering improvements are adopted to demonstrate the feasibility of pre-training with small models.

**Questions To Authors:**

- while i appreciate the effort to release the code, i wonder if there's concrete plan to make the code release more impactful. do you foresee any difficulties of others actually adopting/reusing your codebase for pre-training experimentation? Do you have convergence result for these implementations?

**Reasons To Accept:**

- This is an important question for the academic community to study and benchmark, due to limited resources. Pre-training remains a critical step for model training and the amount of computation required is sometimes a blocker for research ideas to be implemented and experimented in practice
- a good amount of work in put into careful implementation, across a number of different types of models and model sizes. The amount of precious resources put into this work is also large (2k+ GPU hours)
- result with the engineering techniques shows significant speedup compared to the baseline.
- code is available

**Reasons To Reject:**

- only single node experiments are conducted. for large models especially LLMs, multiple nodes may be necessary to train a descent sized model with reasonable amount of time. The networking bandwidth may affect the training throughput. A few reference multi-node result would make this work more comprehensive.
- In the method of empirically estimating the training time, the maximum batch size estimation needs to be further explained as it couples with number of GPUs and memory saving techniques being applied. For instance, with model sharding is applied, the more GPUs are used, the larger batch size can be used for training without OOM. On the other hand, activation checkpointing leads to duplicated computation but saves memory, leading to potentially large batch size which may help reduce communication cost. The principle to select the best optimization techniques and largest batch size remain unclear / over-simplified.
- novelty is a major concern. most of the experiments and implementation are engineering work, lacking research novelty. It is also unclear what is the main challenges
- training time estimation has been studied thoroughly recently, such as a few work mentioned by Echo (https://arxiv.org/pdf/2412.12487). They should have been discussed in this paper why they are not sufficient to answer the question this work bring up

---

> ### Author Response · Authors · 2025-05-30
>
> Dear [DTxJ], thank you for your thoughtful and detailed review! We respond to your concerns below:
>
> - We intentionally exclude multi-node settings from our search space (L84-87; 316-317), because our survey clearly indicated that most academics do not have access to multiple nodes. (We similarly don’t have the multi-node resources to dedicate to this experiment.) But this analysis would make for great future work; for those with resources, our methodology/codebase can be easily extended for this setting.
> - We describe our method for measuring the maximum batch size in L154-157. We will do our best to make this even more clear. You are correct that the maximum batch size is a function of the other settings (the model, GPUs, and training methods) and so we measure this independently for each combination of settings. To do so, we compute a training step with batch size = 1, 2, 4, … (consuming increasing amounts of GPU memory) until we run out of memory and consequently identify this maximum.
> - We address your comment about novelty in [our general comment](https://openreview.net/forum?id=EFxC34XbDh&noteId=xTpguwdEAr). Regarding the emphasis on engineering work, we’d also note that “engineering” is strictly invited in COLM’s [call for papers](https://colmweb.org/cfp.html) (likewise “technological impact” in [reviewing guidelines](https://colmweb.org/ReviewGuide.html)).
> - Thanks for bringing Echo (and the other works in training simulation it refers to) to our attention! We would be happy to refer to these in our paper as well. In general, it seems that simulating training step time (by modeling primitive operations) is far more complex than taking a measurement (as we do); hence, that seems to be an open area of research/engineering. Our approach is both accurate and relatively inexpensive for our purposes and scale. It also permits our framework to be extensible (to arbitrary new training methods; and so that our measurement is not tied to any low-level implementation). We continue to believe that “measuring training step time” is a good design decision for our purposes.
> - We will try to make it as simple as possible to adopt the codebase (and provide clearer documentation, etc). In general, our framework is an abstraction over HuggingFace Transformers, so we can easily inherit models and all training methods with minimal implementation (or glue) on the side of our framework. We have recently added a standalone training script (using the discovered configuration from our benchmarking) into the codebase. Thus, an end-user could (1) test their desired model in our codebase and then (2) pre-train their model in their own project/codebase (by migrating the training script). Their pre-training should thus be no more difficult than doing pre-training with HuggingFace Transformers. We do have ongoing projects that indeed pre-train models using our framework, but will release these at a later time.

---

> > ### Comment · Reviewer_DTxJ · 2025-06-09
> >
> > Thanks for the comments. I have some followup comments regarding approaching being "relatively inexpensive" and the cost of reusing the optimizations in your library.
> >
> > I agree your approach presents an accurate overview of the models included in the paper so far. However, I am not convinced that your approach is inexpensive. The fact that the framework requires end-to-end training on GPUs, even with a few steps, introduce cost whenever a new model architecture/size or GPU count should be considered. With simulators the estimation can be done with just minimum/no GPU execution. It's also unclear how much effort it takes for other existing training codebase to adapt to your library and reuse the optimizations you've built. In general pytorch is a quite flexible library, there're multiple ways to express the training loop (pytorch lightning, pytorch lightning fabric, hf trainer, hf accelerate, etc), let alone hand written training loops like the ones in https://github.com/ByteDance-Seed/VeOmni (which is more researcher-friendly). Could you share the design principle of the optimization APIs, and compare with existing work, to demonstrate the engineering value of your work?

---

> > > ### Author Response · Authors · 2025-06-09
> > >
> > > Thanks for following up!
> > >
> > > 1. Our approach constitutes "a few hours per model–GPU setting" (L241-243; 2 hours on average). If a researcher is planning to dedicate (say) 7–30 days to pre-train a model on some GPUs, it seems very reasonable to spend 2 additional hours (0.1–1%) on benchmarking. Especially when this benchmarking results in a 40% average reduction in training time (Fig. 4; optimal vs free lunch). This seems like a very good return on investment!
> > >
> > > 2. Simulating training would indeed be a more efficient way to estimate training time; in principle, this could reduce the estimation time from hours to minutes. We would be happy to consider these methods in future iterations of our work. That said, existing training simulators are simply unsuitable for our needs. [Echo (Dec. 2024)](https://arxiv.org/pdf/2412.12487) is the latest work in the line of eight simulators (Echo, Table 1). On the upside, it seems to only need 1 GPU and is very fast. However, it does not support Zero-based parallelism (Echo, Sec. 9) and it is unclear which other optimizations from our framework it does support. It also claims 8% error compared to measured training times. Finally, their code has not been made available. Echo also claims (Sec. 8.5) that the prior state-of-art methods have error rates of 20-40%, and require the full cluster of GPUs and take 92x as long in simulation (perhaps on par with the cost of our approach). In all, we continue to believe our design decision is both simple and the best at this time. If an accurate, low-cost, and model/optimization-agnostic simulation method is made available, we will be happy to consider it then.
> > >
> > > 3. Please refer to "choice of software" in Appendix D. We rely on the `transformers.Trainer` API for our training loop. This offers us the advantages of easily supporting the models from and any training optimization made available in the Transformers library. Transformers offers the most complete training library (in terms of available optimizations; compared to PyTorch Lightning, Mosaic ML Composer, etc) and is maintained by HuggingFace. Our engineering effort focuses on: implementing our framework and search space using this training API; our model specifications; our benchmarking procedure; and miscellaneous features to make experiments efficient (better process isolation) and the pipeline complete (CLI, SLURM support, result caching, etc). So our codebase supports our experimental framework and extensibility, but we intentionally defer the implementation of the training loop and optimizations to downstream libraries.

---

### Official Review · Reviewer_Ja5N · 2025-05-12

**Rating:** 7
**Confidence:** 3
**Ethics Flag:** 1

**Summary:**

This paper provides a benchmark to measure the pre-training/training times of different LLM architectures on academic GPU environments.  This work also conducts a survey to define/estimate resources available at an academic level ( around 1-8 GPUs). The main contributions include an overview of LLM architectures reproducible using academic resources and architectures requiring additional hardware/budget. Different optimizations are also explored and covered in this paper to take full advantage of the constrained setups.

Results show most architectures (up to ~1B parameters) can be pre-trained using academic resources. Architectures larger than 2B might not be reproducible by the average resources described in this paper. The architectures covered in the paper include models such as Pythia, RoBERTa, Mamba, ConvNeXt, and ViT.

**Reasons To Accept:**

This paper provides a great overview of different academic setups/resources and techniques to use when pre-training on specific LLM architectures. The results also show the limits of academic resources (i.e. what architecture sizes can't be trained on constrained setups). The hardware/experiment costs shown in the paper also provide useful feedback for labs/researchers writing future proposals on hardware covered by this paper. While these results might not be relevant to researchers in industry or labs with "unlimited" resources, this paper is a great resource for academics who need guidance on budget and feasibility for specific experiments.

**Reasons To Reject:**

While this paper provides a great overview of the architectures trainable on academic resources, this work might become outdated once the next wave of GPUs (multi-core processors) becomes available (or a new wave of smaller LMs).

---

> ### Author Response · Authors · 2025-05-30
>
> Dear [Ja5N], thanks for your review and comments! We’d just note that:
>
> - Our benchmark and framework are designed such that they can be expanded to new GPUs, models, and training methods. Accordingly, this work can be continually updated to reflect the current state of models and resources. We will do our best to do so and this can also be a community effort.

---

### Official Review · Reviewer_ufEP · 2025-05-13

**Rating:** 6
**Confidence:** 4
**Ethics Flag:** 1

**Summary:**

The paper fundamentally tries to answer the question: how long does it take to train an ML model in academia?
First, the paper surveys the hardware resources commonly available for machine learning training in academic institutions. Second, it surveys SOTA DNN training optimizations, classifying them into memory-saving and "free-lunch" techniques. Third, it introduces an automated framework to explore combinations of these optimization techniques to identify the most efficient training approach. Fourth, it presents the results, and compares training times across various models, optimization strategies, and hardware platforms. Fifth, it offers a cost-benefit analysis of different hardware options, enabling academic institutions to make informed decisions while purchasing GPUs.

**Reasons To Accept:**

- Paper is well written.
- I enjoyed reading the paper, and found many insights and information shared interesting.
- Automated framework for parsing through all optimizations is useful for practitioners.

**Reasons To Reject:**

- Technical contributions are limited.
- Empirical estimation of training times is not novel, and indeed common amongst practitioners.
- The paper seems to give the impression that academicians can develop LLMs from scratch,  simply by using fewer GPUs and for longer. However, developing LLMs is a more sophisticated endeavor than that, requiring multiple runs to tune hyper parameters.  This would mean academicians need access to limited GPUs for months at stretch to make significant progress, currently not typically available to academicians.
- There is little description of how the framework makes support of adding new optimization techniques easy and amenable.  This is crucial for long term applicability of the automation framework, given the pace of innovation in ML and new training optimization techniques.

---

> ### Author Response · Authors · 2025-05-30
>
> Dear [ufEP], thank you for your thoughtful review! We are glad you enjoyed reading our paper. Please see our responses below:
>
> - We respond to your note about technical contributions and methodological novelty in [our general comment](https://openreview.net/forum?id=EFxC34XbDh&noteId=xTpguwdEAr).
>
> - We would like to clarify that our intent is not necessarily to enable academics to train competitive LLMs from scratch. Instead, we want to provide clarity and transparency about what is feasible in academia. Such information is currently not readily available to most academics. That said, we do see some promising research avenues that are available and likely overlooked. E.g. academics may begin conducting controlled studies involving pre-training (smaller) LLMs. I.e. to measure the causal effects of individual variables during pre-training (a scientific practice that is strongly neglected in this area). Such studies could include: testing new training objectives (as in the CLIP example; L24-30), data diets, architectures, optimizers, and so forth. Of course, all experiments require some hyper-parameter tuning, but it should be possible to estimate the resources for these additional runs as well (using our framework) and design studies with this budget in mind. Again, the idea is for academics to estimate what is feasible: a consequence of that will be that there are indeed some infeasible experiments. But some are feasible, and we’d like academics to be in a position to tell the difference.
>
> - You’re correct. We claim that our framework can be easily expanded, but avoided describing this in detail in the paper body. We will definitely elaborate further in revisions. In general, our framework is an abstraction over HuggingFace Transformers. Thus, we can inherit any training method they support. To add new methods (e.g. from new versions of Transformers) into our framework, one must edit `src/train.py` and `experiments/config.py`, which jointly map our framework’s search space parameters to `Trainer` and `TrainingArguments` from `transformers`. New models can also be added by implementing the abstract model classes in `src/models/__init__.py`. We believe these are relatively easy interventions and we will work to make these even easier. These changes automatically propagate into our experiment runners and CLIs. In newer iterations, we also provide a standalone training script (receiving configurations identified by our benchmarking), so that end-users can use our codebase for testing and easily migrate the pre-training experiment to their own projects.

---

> ### Comment · Reviewer_ufEP · 2025-06-08
>
> - I appreciate the overview of your codebase. However, it is still challenging to evaluate the effort required to add new optimizations.  An acceptable, albeit imperfect, metric to quantify this effort could be lines of code.  For instance, if you were to add a new optimization not included in your codebase, such as pipeline parallelism (supported by DeepSpeed), how many lines of code would need to be added/modified?
>
>     Many optimizations also require tuning several knobs to achieve optimal performance. Continuing with the pipeline parallelism example, performance can depend on parameters such as number of pipeline stages, micro-batch size and scheduling strategy. How does the proposed automation framework support tuning such parameters, and including them in the search space?
>
> - The addition of new optimizations causes the search space to grow factorially.  Evaluating combinations of 5 optimizations (activation checkpointing, ZeRO stages 0, 1, 2 and offloading) require “a few hours per model-GPU setting”. Adding even a couple more optimizations can likely require multiple days for an exhaustive optimal search.  Could you comment on the feasibility of the  automation framework with growing optimizations?
>
>
> - I appreciate the quantification of net improvements observed using optimal combination of memory-saving techniques.  However, it would be helpful to know which specific combinations proved optimal.  Could you share, for instance, the combination that performed best for the Pythia 160M model on 8 A100 GPUs?  Which combination performed the "worst"?
>
>   Could you also comment on whether a reasonably experienced ML researcher could have identified these optimal (or near-optimal) combinations without the aid of your automation framework? I understand that “experience” is subjective, but any insights would be valuable.
>
>     Separately, publishing the optimal combinations you found for different models would be immediately useful to the ML community.  Open sourcing this information, building an organized and searchable database, and encouraging academicians to contribute to this database would be highly impactful and serve as a long term solution.

---

> > ### Author Response · Authors · 2025-06-09
> >
> > Thanks for the very thoughtful follow-up.
> >
> > 1. Our code is a wrapper over `transformers.Trainer`. Thus, adding new optimizations to our library is no more difficult than to Transformers/DeepSpeed-based training code.
> >     - Let's consider two newly-added optimizations. One can add [Liger Kernels](https://arxiv.org/abs/2410.10989) primarily with: `TrainingArguments(use_liger_kernels={True, False})` (`src/train.py:L122`). You'll need 4 more assignments to add the boolean choice to our config classes and propagate it to the `TrainingArguments`. One can also set a model to use [Flex Attention](https://arxiv.org/abs/2412.05496) just by switching `attn_implementation` from `"sdpa"` to `"flex_attention"` (e.g. `src/models/pythia.py:L37`).
> >     - For more involved optimizations (sharding/offloading): our codebase dedicates ~100 lines of code for integrating DeepSpeed and ~30 lines of code for FSDP.
> >     - Implementing pipeline parallelism is the exact same as if a user was using DeepSpeed alone. DeepSpeed [requires users to re-implement models](https://www.deepspeed.ai/tutorials/pipeline) and so a user would add that implementation to `src/models/` in our codebase (length subject to their model/implementation of choice).
> >
> > 2. Yes, some optimization methods can be further tuned. We generally opted to use default settings. Interested users can extend the search space to consider these; as we describe in (1), they must simply add additional fields to our config class and propagate those values to the `TrainingArguments`.
> >
> > 3. To clarify, the search space permutes: checkpointing [2], sharding [6], and offloading [2] (number of choices in brackets). Adding new dimensions would indeed cause the space to grow combinatorially. But, non-orthogonal methods grow the space by a smaller fraction. Suppose a new method is incompatible with the existing sharding methods (e.g. because it is a new sharding method): then we will have 7 choices in the sharding dimension and growth by 1/6. Finally, adding new "free lunch" methods induces no growth to the search space.
> >     - In general, there are not that many "new dimensions" of memory-saving methods. The three listed dimensions cover all (model-agnostic) methods since 2019. The two prominent optimizations introduced in the last year [mentioned in (1)] are both "free-lunch" methods.
> >
> > 4. We provide a table of optimal configurations in Appendix J. Specifically, the best configuration (2 days) for Pythia (160M) on 8 A100 GPUs involves using the "free lunch" methods and Zero stage 1 sharding. The worst configuration (9 days) is the "naive" setting; the second worst configuration (5 days) uses "free lunch" methods, activation checkpointing, Zero stage 3 sharding, and offloading. Since this is the "small model / large GPU" regime.
> >
> > 5. We feel that an experienced (or even expert) ML researcher would struggle to identify a near-optimal configuration without some trial-and-error. The [general](https://huggingface.co/docs/transformers/v4.39.2/perf_train_gpu_one) [recommendation](https://huggingface.co/docs/transformers/v4.39.2/perf_train_gpu_many) is to test one optimization at a time, using some intuitions. Our framework automates this, is more thorough, and lowers the barrier for expertise.
> >     - While experts may have good intuitions about each of the optimizations independently, the interaction of these optimizations (esp. compilation, kernels, sharding, etc) becomes very complex and is further nuanced by chosen models and GPUs: the exact time–memory tradeoff becomes very difficult to anticipate. Moreover, the threshold for GPU memory consumption is a very thin line; training will simply fail if one at all exceeds this limit.
> >     - In Figure 4 (only considering configurations that did not over-allocate memory), we show that the optimal configuration in the search space is 2.34 times faster than the median configuration and (not shown) 1.26 times faster than the second-best. Picking an optimal configuration (of 22 choices) is non-trivial.
> >
> > 6. We completely agree that publishing the optimal configurations from our benchmarks would be valuable. We intend to fully open-source our work and, as you are suggesting, it would also be great to collect contributions from the community as well.

---

### Author Response · Authors · 2025-05-30

Dear reviewers: there seems to be a general concern about “novelty” (i.e. a lack of newly proposed methods) in our work. This is intentional; our aim is solely to answer our research question and, in doing so, we believe our work is well-aligned with [COLM’s listed dimensions of valued contributions](https://colmweb.org/ReviewGuide.html) (bolded as follows). We try to conduct the necessary analyses (following **principled approaches**) to answer this question. We simultaneously aim to provide **technical resources / empirical artifacts** to our community; enable new research paradigms for our community (**forward outlook**); and generally present our findings with openness (**clarity/honesty**). We believe our research question and tailored methods (e.g. survey, framework + search space, analyses) are what make our work original. We thank you for your consideration and open-mindedness.

---

### Decision · Program_Chairs · 2025-07-08

**Decision:**

Accept

**Comment:**

This paper addresses a critical practical question for the academic community by systematically investigating whether researchers with limited compute resources can feasibly pre-train large language models. Through comprehensive surveys of academic GPU availability and extensive empirical benchmarking across 2,000 GPU-hours, the authors demonstrate that meaningful pre-training is more accessible than commonly assumed—showing, for example, that Pythia-1B can be replicated on 4 GPUs in 18 days rather than requiring 64 GPUs for 3 days. The work provides valuable practical insights through automated optimization framework exploration and cost-benefit analyses that will directly benefit resource-constrained academic researchers. While the technical contributions are primarily engineering-focused rather than algorithmically novel, the systematic nature of the study, substantial experimental investment, and immediate practical utility to the academic community make this a valuable contribution that fills an important gap in understanding training feasibility constraints.

Pros: Addresses crucial practical needs of academic researchers with limited compute; comprehensive empirical study with substantial experimental investment (2,000+ GPU-hours); automated framework for exploring optimization combinations; valuable cost-benefit analysis for hardware procurement decisions; well-written and accessible presentation; code release enhances reproducibility and community impact.

Cons: Oversimplified view of LLM development that doesn't account for hyperparameter tuning requiring multiple training runs; unclear principles for selecting optimal batch sizes and optimization techniques; missing discussion of recent related work on training time estimation. The work could be improved with more detailed analysis of computational efficiency metrics like FLOPs/sec beyond wall-clock time measurements.